# MQuad enables clonal substructure discovery using single cell mitochondrial variants

Aaron Wing Cheung Kwok [1], Chen Qiao [1], Rongting Huang [1], Mai-Har Sham[2], Joshua W. K. Ho [1,3 ✉] & Yuanhua Huang [1,4 ✉]

Mitochondrial mutations are increasingly recognised as informative endogenous genetic markers that can be used to reconstruct cellular clonal structure using single-cell RNA or DNA sequencing data. However, identifying informative mtDNA variants in noisy and sparse single-cell sequencing data is still challenging with few computation methods available. Here we present an open source computational tool MQuad that accurately calls clonally informative mtDNA variants in a population of single cells, and an analysis suite for complete clonality inference, based on single cell RNA, DNA or ATAC sequencing data. Through a variety of simulated and experimental single cell sequencing data, we showed that MQuad can identify mitochondrial variants with both high sensitivity and specificity, outperforming existing methods by a large extent. Furthermore, we demonstrate its wide applicability in different single cell sequencing protocols, particularly in complementing single-nucleotide and copy-number variations to extract finer clonal resolution.

[1] School of Biomedical Sciences, Li Ka Shing Faculty of Medicine, The University of Hong Kong, Pokfulam, Hong Kong SAR, China. [2] School of Biomedical Sciences, Faculty of Medicine, The Chinese University of Hong Kong, Shatin, Hong Kong SAR, China. [3] Laboratory of Data Discovery for Health Limited (D24H), Hong Kong Science Park, Hong Kong SAR, China. [4] Department of Statistics and Actuarial Science, The University of Hong Kong, Pokfulam, Hong Kong SAR, China. ✉email: jwkho@hku.hk; yuanhua@hku.hk

dentification of clonal relationships in a population of single cells is a major challenge of single cell data science[1]. Such information is essential in recovering cell lineages, which can have broad applications in developmental[2,3], stem cell[4] and cancer biology[5]. In particular, deciphering intra-tumor genetic heterogeneity and clonal mutations are cricitial for revealing their evolutionary dynamics and drug resistance of cancers[6]. Fortunately, recent advances in single-cell sequencing bring promises to the identification of subclonal structure in tumors[7,8] and characterization of phenotypic impacts from single nucleotide variations (SNVs) and copy number variations (CNVs)[9,10]. However, SNV-based lineage reconstruction remains a grand challenge[1] partly due to the relatively small number of somatic variants in the large nuclear genome[11] and a high dropout rate[12], especially for assays with low efficiency and limited coverage, e.g., Smart-seq2 for transcriptome[13]. CNVs inferred based on single cell transcriptomic data have been widely used[14,15], but subclonal structures inferred from read-coverage-inferred CNVs might not be as dynamic as subclones shown by the propagation of true point mutations. It has been shown recently that mitochondrial heteroplasmy serves as an excellent alternative to nuclear SNVs for studying lineages[4,16–18] or human embryogenesis and development[2,3] due to the mitochondria's large number of copies and higher mutation rate (>10 fold of nuclear genome[19]). Thanks to its cost efficiency and high accessibility, mtDNA variations have since attracted great attention, with further development of novel droplet-based single-cell sequencing protocols to enrich mitochondrial sequences in a highly multiplexed manner, such as mtscATAC-seq[16].

However, it is highly challenging to differentiate between clone-discriminative mtDNA mutations and non-inherited mutations that are totally unrelated to the lineage structure. Very few computational methods are available for analyzing mitochondrial SNVs across different sequencing assays, especially for conventional single-cell RNA sequencing (scRNA-seq) data. Nuclear SNV callers such as Monovar[20] and Conbase[21], assume a diploid context which is violated in the mitochondrial genome. MtDNA specific methods, EMBLEM[22] and mgatk[16] were designed primarily for single-cell assay for transposase-accessible chromatin with sequencing (scATAC-seq) and the SNV quality detected in other data types such as scRNA-seq remains to be further evaluated. Combining with the long-standing problem of sequencing errors, uninformative and noisy mtDNA SNVs greatly confound clonal inference and biological interpretation.

To address these limitations, we develop a computational method called MQuad (Mixture Modeling of Mitochondrial Mutations, M[4]) that effectively identifies informative mtDNA variants in single-cell sequencing data for clonality inference. Importantly, MQuad can be used in combination with two other recently developed tools to form an integrated clonality discovery pipeline, cellSNP-MQuad-VireoSNP, which provides a complete analysis suite from single cell mtDNA genotyping to clonal reconstruction. We demonstrate its usage on various single-cell sequencing data sets to identify clones based on mtDNA mutations. More importantly, our analysis reveals that mtDNA mutations detected by MQuad can be used in complement with nuclear SNVs and CNVs to achieve finer clonal resolution.

## Results

### MQuad is a robust statistical approach to identify informative mtDNA variants.
MQuad is a computational method for detecting clone discriminative mitochondrial variants. It is tailored to work seamlessly with cellsnp-lite[23] and vireoSNP[24] to create an automated end-to-end pipeline for single cell clonal discovery using mitochondrial variants (Fig. 1a). Briefly, MQuad fits the alternative and reference allele counts of each variant to a binomial distribution with either one shared parameter (i.e., the expected alternative allele frequency) across all cells under the null hypothesis $H_0$ or two different parameters in the cell population as the alternative hypothesis $H_1$ (i.e., two-component mixture; Methods). Instead of assuming a diploid context like in the nuclear genome, the binomial parameter(s) here can range from 0 to 1 for different levels of heteroplasmy. The difference of the Bayesian Information Criterion scores of the fitted $H_0$ or $H_1$ models ($\Delta BIC = BIC(H_0) — BIC(H_1)$) is then used to prioritize the candidate variants that are informative with respect to clonal discovery, with a higher $\Delta BIC$ for stronger support of the alternative model that the variant is clonally informative. Then, a cutoff on $\Delta BIC$ is determined automatically by the inflection point (a knee point) in the cumulative distribution of $\Delta BIC$ (Methods). With a highly discriminative set of mtDNA variants identified by MQuad, vireoSNP clusters single cells to clones based on their mtDNA mutation profiles.

We first benchmarked MQuad with simulated data that mimicked scRNA-seq data generated from the Smart-seq2 protocol. To simulate three clonal populations, we separated the dataset into three groups of cells and designated 5-50 clone-specific mtDNA variants for each group (Supplementary Fig. 1 and Methods). On top of that, spontaneous mutations and sequencing errors were generated at a low frequency across all cells to simulate a noisy background (Methods).

We compared MQuad's performance against two other single-cell variant callers: mgatk[16] and Monovar[20]. In an example simulation with 50 variants per clone and all other default settings (Methods; Fig. 1), MQuad has the best overall performance in identifying ground truth clonal variants (area under precision recall curve AUPRC = 0.976, Fig. 1b; area under receiver operator characteristic curve AUROC = 1.00, Fig. 1c), outperforming both mgatk (AUPRC = 0.800, AUROC = 0.999) and Monovar (AUPRC = 0.147, AUROC = 0.968). All tested tools have high AUROC (>0.95) because there is an imbalance between the number of clonal variants (15-150) and the number of true negative variants (>16,000). The precision recall curve shows that MQuad has a better control in false positive SNPs than other tools, which demonstrates that it is non-trivial to detect mitochondrial variants precisely. Monovar was designed for detecting somatic variants in the nuclear genome instead of the mitochondrial genome. The violation of diploid assumption is probably the reason that Monovar poorly distinguishes technical noises hence returning many false positives. For mgatk, we reason that the variance mean ratio (VMR) used by the algorithm is difficult to estimate correctly and may suffer from high uncertainty, hence is not a robust predictor of informativeness in scRNA-seq due to an abundance of low allele frequency variants and sequencing errors. In contrast, our proposed metric, $\Delta BIC$, was sensitive to distinguish between informative variants and noise, which made it more intuitive to place a cutoff at the knee point with the sharpest increase of $\Delta BIC$ (Fig. 1d; circle on blue PR curve in Fig. 1b, Recall = 0.93, Precision = 0.79).

We also varied several key simulation parameters to explore the effect of dataset characteristics on clonal variant discovery, including the number of informative variants per clone, the allele frequency of clonal variants, the ratio of clone sizes, and evolutionary models (Fig. 1e–h). Across almost all settings, we found MQuad outperforms mgatk and Monovar by large gains in AUPRC, suggesting its enhanced performance in controlling false positives. The only exception is when the allele frequency of clonal variants is at or lower than 1%, in which case all tools perform poorly (Fig. 1f). This is not surprising as such low allele frequency is close to the level of technical noise (average allele

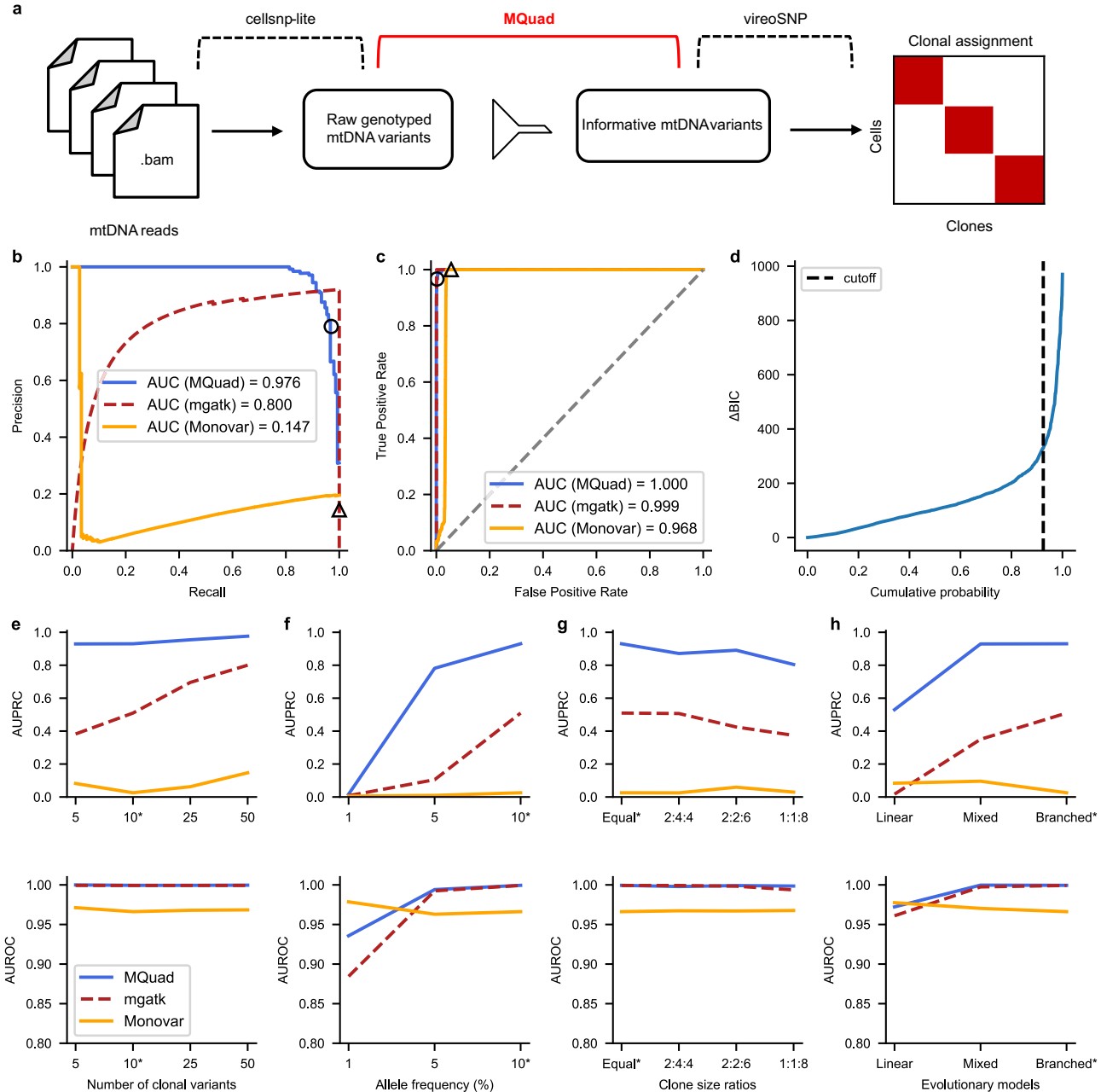

**Fig. 1 Overview of analysis pipeline and benchmark with simulated data. a** Schematic for tailored analysis suite recommended to use alongside MQuad. Precision-Recall (PR) curve (**b**) and Receiver Operating Characteristic (ROC) curve (**c**) for the detection of simulated variants in a simulation with all default settings (see Methods) except 50 clonal variants per clone. Curve is generated by varying the cutoff on ΔBIC in MQuad, VMR in mgatk, and MPR in Monovar. Black circle and triangle represent the default thresholds used to classify informative variants for MQuad and mgatk respectively. The threshold for Monovar is not shown because there is no default threshold suggested. **d** Cumulative distribution function of ΔBIC with cutoff shown, based on the same data in **b**, **c**. Changes in area under PR curve (AUPRC) and area under ROC curve (AUROC) when varying (**e**) number of variants per clone, (**f**) allele frequency of clonal variants, (**g**) clone sizes, (**h**) evolutionary models. Asterisk indicates the default parameter.

frequency: 0.44%; Supplementary Fig. 1a). We also observed that variants propagated through a linear evolution model are the hardest to detect for all tools, since the clone sizes are usually skewed and most variants are shared between multiple clones.

**Tumor cell populations are accurately inferred from informative mtDNA variants detected by MQuad.** Next, we applied MQuad to a clear cell renal cell carcinoma (ccRCC) scRNA-seq dataset[25] that contained a mixture of with three distinct source populations of cells from the same patient: patient-derived

xenograft of the primary tumor (PDX pRCC); patient-derived xenograft of metastatic tumor (PDX mRCC); and metastatic tumor directly from the patient (Pt mRCC).

MQuad detected 146 informative mtDNA variants from the dataset. We observed heterogeneity in allele frequencies of these variants with some being highly clonal-specific, e.g., 7207G>A was mostly specific to PDX pRCC alone and not found in the metastatic tumor populations (Fig. 2a). We also observed a sharp increase in the cumulative distribution of ΔBIC (Fig. 2b) which was consistent with simulated data, further justifying the rationale behind determining the cutoff based on a knee point.

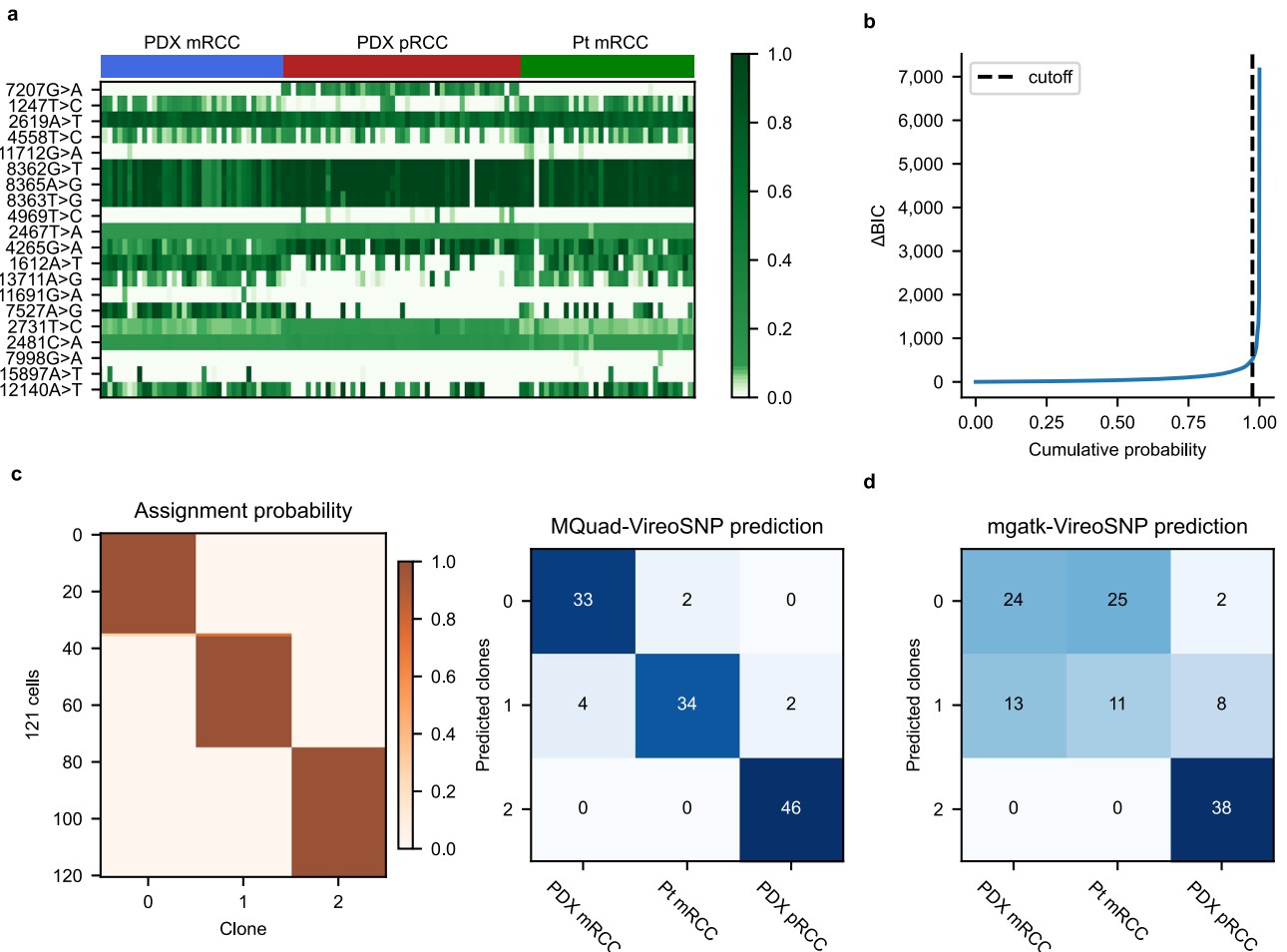

**Fig. 2 Distinct tumor populations inferred from heterogenous mtDNA variants. a** Allele frequency heatmap showing the top 20 informative mtDNA SNVs detected by MQuad ranked from highest to lowest ΔBIC. Each row is a variant, each column is a cell. **b** Cumulative distribution function of ΔBIC with cutoff shown. **c** (Left) Clonal assignment with mtDNA variants detected with MQuad. Each row is a cell, each column is a clone, heatmap color indicates assignment probability. (Right) Confusion matrix between predicted mitochondrial clones and source labels. Numbers represent cells assigned. **d** Confusion matrix between predicted clones based on mgatk variants and source labels.

To further validate the accuracy of MQuad, we used the detected variants as an input to vireoSNP for clonal assignment. We found that most cells were confidently assigned to their respective origins with high concordance (Fig. 2c). Our assignment achieved 93% concordance with the source labels, which is consistent with earlier reports on this dataset that cells from different sources display distinct copy number variations[25] and nuclear SNVs[26]. Therefore, it implies that genetic heterogeneity in the mitochondrial genome can indeed be leveraged for reconstruction of tumor clonal subpopulations. Admittedly, the source labels here are only a coarse annotation of the tumor clonality, as it is generally challenging to obtain a detailed ground truth.

We also assigned clones based on variants detected by mgatk (Supplementary Fig. 3a). While mgatk detected more than double the number of variants (312 mtSNVs), the clonal assignment was less concordant (64%) with the source labels (Fig. 2d). The two metastatic tumor populations were less distinguishable from each other, possibly due to a large number of false positive mtSNVs that mgatk failed to filter out.

**MQuad identifies mtDNA-based clonal structure that complements clones inferred from nuclear SNVs.** We further applied MQuad to a recent scRNA-seq dataset that characterized

the somatic clones in healthy fibroblast cell lines[9]. The fibroblast cell line used, joxm, was from a white female aged 45-49. We observed even in cells that were not known to be tumorigenic, a substantial level of mitochondrial genetic heterogeneity could be detected. MQuad detected 24 SNVs with which we assigned the 77 cells into 3 clones (Fig. 3a). Not only did we observe distinct mutations in specific clones (e.g., 11196G>A, Fig. 3a and Supplementary Fig. 4), we also identified random genetic drift events that lead to different heteroplasmy levels between clones (e.g., 2619A>T, Fig. 3a and Supplementary Fig. 4).

Comparing our clonal assignment to the clones inferred from nuclear mutations, mtDNA clones showed partial but consistent concordance with nuclear clones (Fig. 3b). In particular, nuclear clone 3 was completely identical to mitochondrial clone 2, while nuclear clones 1 and 2 were more ambiguous in terms of mtDNA. Using our MQuad-identified mtDNA variants, our pipeline discovered two subclones MT0 and MT1 within nuclear clone 1 that had distinct mtDNA mutation profiles (Fig. 3c). Next, we tested if MT0 and MT1 could indeed be biologically and clonally distinct cell populations. Differential expression (DE) analysis between cells from the two subclones of nuclear clone 1 (clone1.MT0 & clone1.MT1) identified 847 DE genes (two-sided edgeR QL F-test; FDR < 0.1; Fig. 3d). A large number of highly expressed DE genes in clone1.MT0 were enriched for cell proliferation[27,28] (gene sets MYC targets, E2F targets, G2M

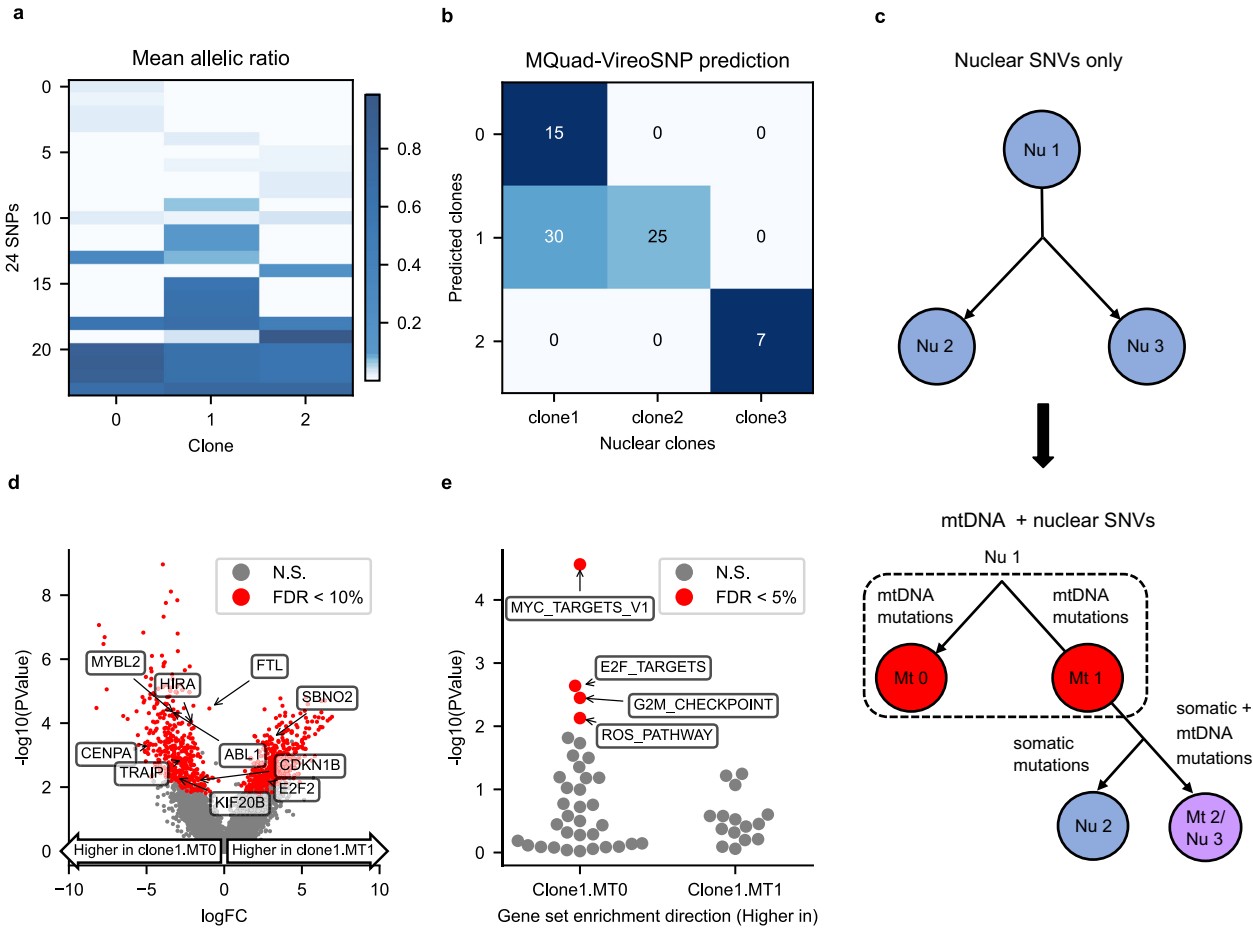

**Fig. 3 Combination of mtDNA and nuclear SNVs identifies finer clonal structure in healthy fibroblast. a** Mean allelic frequency of mtDNA variants in each clone. Each row is a mtDNA SNV, each column a clone. **b** Confusion matrix between predicted mtDNA clones from MQuad and predicted nuclear clones from Cardelino. **c** (Top) Clonal tree inferred from nuclear SNVs alone. (Bottom) Clonal tree inferred from both nuclear and mtDNA SNVs. (Nu Nuclear clone, Mt Mitochondrial clone). **d** Volcano plot showing negative log10 *P*-values (edgeR two-sided quasi-likelihood *F*-test) against log2-fold changes (FC) for DE between cells assigned to Nu1 & Mt 0 and Nu1 & Mt 1. Significant DE genes (FDR < 0.1) highlighted in red. **e** Enrichment of MsigDB Hallmark gene sets based on log2 FC between Nu1 & Mt 0 and Nu1 & Mt 1 (two-sided camera test). Gene set enrichment direction is Nu 1 & Mt 1 with respect to Nu 1 & Mt 0. Negative log10 *P*-values of gene set enrichments are shown with significant gene sets (FDR < 0.05) highlighted and labeled.

checkpoint, Reactive Oxygen Species (ROS) Pathway; two-sided camera test; FDR < 0.05; Fig. 3e). The original study which published this dataset identified cells in clone 1 has a higher proliferation rate than clone 2. Using MQuad, our results further discovered that a specific subclone MT0 within clone 1 is likely the main contributor of the elevated proliferation rate.

**MQuad identifies subclonal structure in a gastric cell line with scDNA-seq.** We further asked if mitochondrial mutations could be used in conjunction with copy number variations (CNVs) to infer clonal evolution in tumors. We first examined the applicability of MQuad on a barcode-based single-cell whole genome sequencing (scDNA-seq) dataset, e.g., 10x Genomics, as it had attracted a lot of attention recently owing to its accuracy of identifying CNVs and clonal structure despite its low coverage[29,30]. By re-analyzing a publicly available gastric cancer cell line MKN-45 (scDNA-seq in 10x Genomics) with both cellranger and our own B-allele frequency (BAF) clustering, we identified two distinct CNV clones (Fig. 4a, Supplementary Fig. 3; Methods), which was in agreement with the original analysis[29]. Both clones shared multiple CNVs, e.g., copy loss on chr4p, but

clone 1 represented a small population of cells (7%) having a unique copy loss on chr4q.

By applying MQuad to MKN-45 scDNA-seq, we detected 24 clone discriminative mitochondrial mutations even with a relatively low sequencing depth and were able to infer 5 clones with vireoSNP (Fig. 4c). Comparing the clone assignment to CNV profiles, we observed a high degree of overlap between the CNV clone 1 and mitochondrial clone 0 (Fig. 4b, c). Based on the raw allele frequencies of mtDNA variants (Fig. 4c), we discovered the most distinctive clone of MT0 is marked by the presence of 2393C>T and 8368G>A, which were largely absent in all other clones (Fig. 4c). There are also a few common mutations (either germline or early somatic mutations) with varying VAF detected across clones (4184T>C, 16286C>T, 1841T>C, 15894A>G, 11166G>A, 14552G>A), showing the effect of random genetic drift between clones. As there is no tailored method for linear tree inference based on mtDNA variants, we used SCITE[31], a nuclear SNV-based method, to construct a lineage tree using mtDNA SNVs, then incorporated CNV information into the phylogeny manually (Fig. 4d). It should be noted that SCITE only uses the presence and absence of a mutation (mean AF > 0.01 for presence here) as input. We reason that mtDNA SNVs have a higher

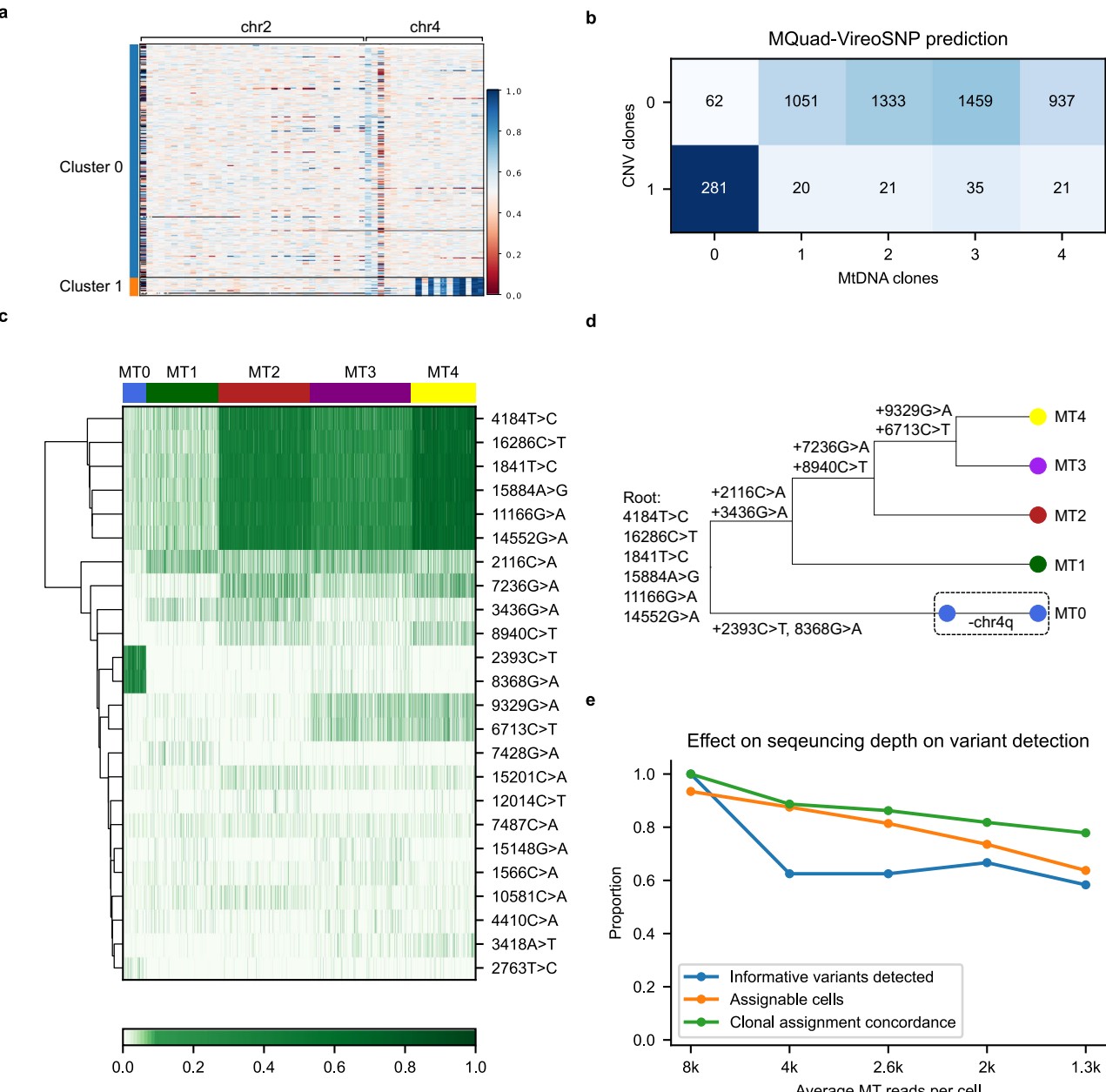

**Fig. 4 Mitochondrial mutations show concordance with CNV in scDNA-seq. a** CNV profile of gastric cancer cell line MKN-45. Shown is B-allele frequency across 5220 cells (in row) on chr2 and chr4 on heterozygous SNPs aggregated in 5MB bins through phasing in a three-step strategy (Methods). **b** Confusion matrix between CNV clones and mitochondrial clones. **c** Allele frequency heatmap on 24 clonally discriminative mtDNA variants detected by MQuad. **d** A speculated lineage tree on CNV and mitochondrial clones considering the CN events and mtDNA allele frequency. **e** Effects of sequencing depth on mtDNA variants identification and cell assignment to clones.

variability than large-scale chromosome level aberrations, thus providing a finer resolution on clones that are not easily distinguishable with CNVs alone.

Next, we downsampled the data into various depths to quantify the effect of sequencing depth on mtDNA variant calling and clonal assignment (Fig. 4e). Compared to the full dataset, we observed a linear drop in number of cells with assignable clones, number of informative mtDNA detected, and clonal assignment concordance as the number of reads per cell decreases. Although MQuad and vireoSNP are robust against missing data, to maximize the proportion of assignable cells, we recommend a minimum of 3,000 mitochondrial reads per cell for applying MQuad to perform adequately on scDNA-seq data, assuming that the reads were distributed evenly along the mitochondrial genome.

**MQuad can detect mtDNA variants near captured sites in UMI-based scRNA-seq.** One of the most widely used platforms for scRNA-seq is droplet-enabled UMI-based scRNA-seq, such as those generated by the 10x Genomics platform. One of the main limitations of using such data for variant calling is that the reads are typically only enriched for the 3′ or 5′ end of a transcript, and hence resulting in reads that have highly non-uniform coverage. To test the performance of MQuad and the clonal assignment pipeline, we evaluated its application on three 3′-biased scRNA-seq datasets generated by the 10x Genomics platform from triple negative breast cancer samples (TNBC1, TNBC2 and TNBC5)[14]. Unsurprisingly, even with a comparable number of mitochondrial reads, 10x scRNA-seq performed significantly worse than other sequencing protocols. Only a small number of mtDNA variants

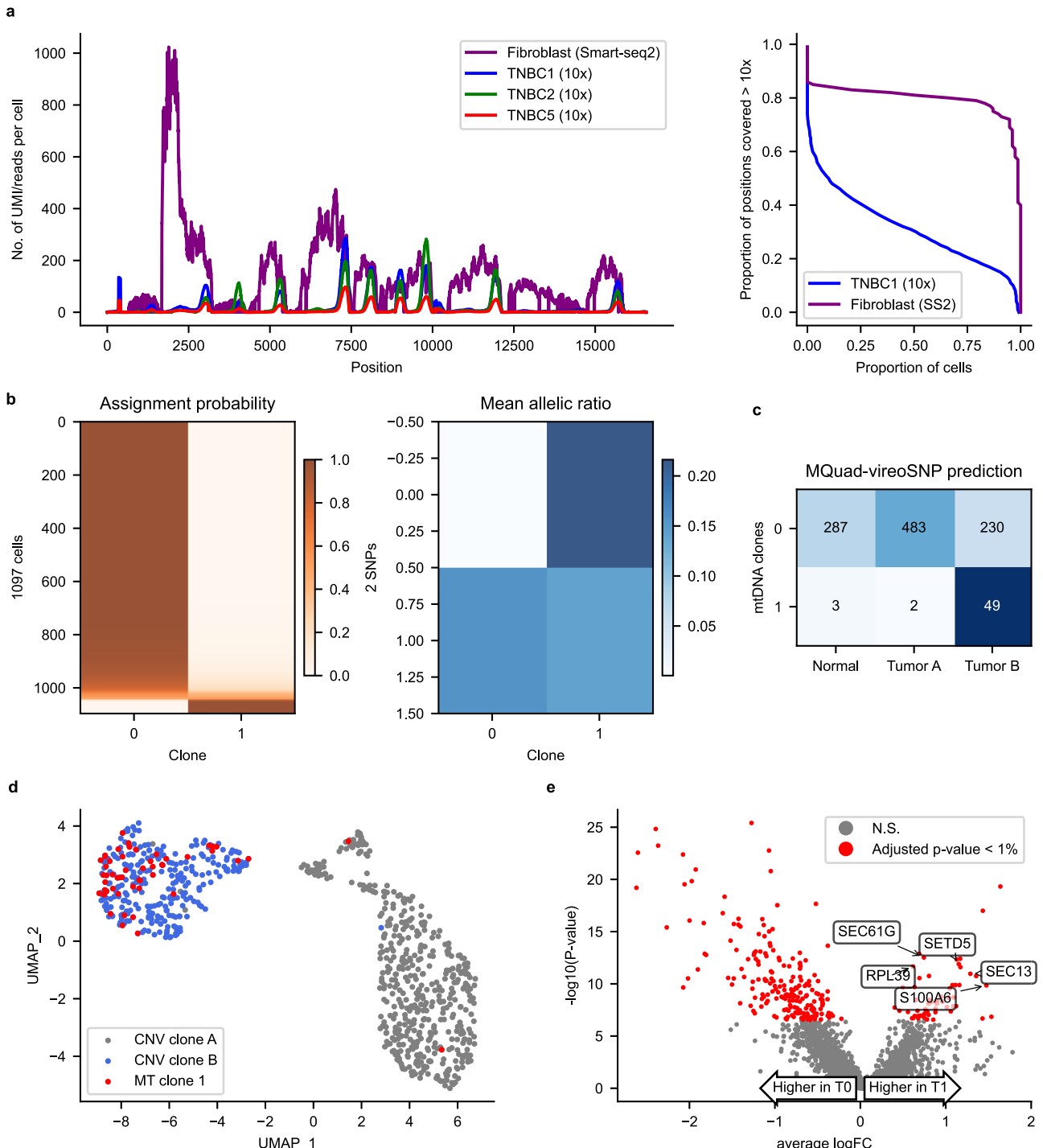

**Fig. 5 Analysis of mtDNA variants on 10x scRNA-seq datasets. a** Coverage comparison between 3 TNBC 10x scRNA-seq datasets and fibroblast Smart-seq2 dataset. UMI-based scRNA-seq shows uneven distribution of reads across the mitochondrial genome while Smart-seq2 is generally more well-covered. **b** Clonal assignment and mean allelic ratio of 1097 cells from TNBC1. **c** Confusion matrix between CNV subclones and confidently assigned mitochondrial clones. **d** UMAP plot with tumor cells only. Mitochondrial clone 1 highly overlaps with CNV clone B. **e** Volcano plot showing negative log10 P-values (Seurat FindMarkers with DESeq2 model; two-sided Wald test) against average log2 fold changes (FC) for DE between T0 (Tumor & MT0) and T1 (Tumor & MT1). Significant DE genes (Adjusted *p*-value < 0.01) highlighted in red. DE genes that are associated with breast cancer are labeled.

were found and the numbers of clones identified were small across all datasets (Supplementary Table 1). Coverage analysis showed this might be due to the highly uneven coverage of mitochondrial reads in this 3′-biased 10x Genomics scRNA-seq dataset (Fig. 5a). No or low number of cells can be confidently assigned to a clone in TNBC2 and TNBC5. Similarly, limited power is observed in a 5′-biased 10x Genomics scRNA-seq data

(Supplementary Table 1), confirming the limited performance due to the uneven read coverage.

Nonetheless, MQuad detected two informative variants in the TNBC1 dataset and assigned 1097 cells (95.8%) with probability >0.8 into two clones (Fig. 5b). In this case, the identified mtDNA variants were located close to the 3′ end of some genes. We found that the small mitochondrial clone 1 almost exclusively (51 out 54

cells) came from the tumor portion and showed a significant enrichment compared to mitochondrial clone 0 ($p = 6 \times 10^{-5}$, Fisher's exact test; Fig. 5c). Moreover, mitochondrial clone 1 highly overlapped with CNV subclone B identified from the original paper[30] (Fig. 5d), indicating the presence of a small subclone within a CNV clone. By examining the differentially expressed genes between the two mtDNA clones, we identified 57 up-regulated genes in subclone 1 (Adjusted $p$-value < 0.01, Seurat FindMarkers function with DESeq2 model; Fig. 5e), many of which are known to be associated with breast cancer, i.e., SEC61G[32] and RPL39[33]. High level of SETD5 expression is also related to poor prognosis in breast cancer[34], providing additional evidence for the biological significance of mitochondrial clone 1.

Taken together, this analysis demonstrated the ability of MQuad to identify informative mtDNA variants to identify biologically meaningful cell subclonal structure, even in 3′ bias 10x Genomics data, suggesting that clonally informative variants can be detected if they are close to the capture sites, which opens up a wide possibility to re-analyze a large volume of UMI-based scRNA-seq data in the public domain.

Moreover, the technical issue of low read count and uneven coverage may be overcome by using emerging protocols for mitochondrial sequence enrichment[35]. As an indirect reference, when applying MQuad to a barcode-based single-cell ATAC-seq (mtscATAC) dataset from colorectal cancer[16], MQuad in general returned strong evidence (high △BIC) for clonal variants and identified a set of clonal variants with high consistency to the original report (Supplementary Fig. 7). Overall, it shows the need for a highly effective computational method for detecting mitochondrial variants like MQuad, ideally with optimized sequencing coverage, and using such variants to detect the fine clonal structure in the cell population.

## Discussion

To date, mitochondrial sequences are often overlooked in single cell sequencing data analysis. We demonstrate here a tailored analytical suite that can harness these mitochondrial sequences to discover clone-discriminative genetic variants using standard single cell sequencing data. Without requiring additional mitochondrial enrichment steps, our pipeline is applicable to most existing single-cell data with a sufficient sequencing depth and even coverage (see Supplementary Table 1 for reference). This means that cell lineage information can be discovered with virtually no additional experimental cost. We make the MQuad source code and tool publicly available, and the program is designed to work seamlessly with other recently published tools, cellsnp-lite[23] and vireoSNP[24]. Our analysis shows that leveraging mtDNA mutations can decipher not only tumor clonal dynamics but also resolve clonal substructure that may not be detectable based on nuclear DNA alone.

The key strength of MQuad is that it adopts a model selection approach in evaluating the informativeness of each variant, which is more robust than considering the raw allele frequencies alone. Especially in deeply sequenced data with a lot of read counts (e.g., Smart-seq2), this approach is more effective in reducing false positives compared to existing methods. With the emergence of massively parallel sequencing protocols that enriches mitochondrial reads, dealing with noise will be inevitable and MQuad serves as a flexible option to filter for useful mtDNA variants. Here we show that MQuad can be applied to sequencing data of various nature and we anticipate that it can adapt well to future sequencing technologies and datasets.

With the possibilities enabled by mtDNA lineage reconstruction, dealing with different types of genomic alterations occurring in the same cell remains an open challenge and there is a high demand for effective computational methods for this purpose. In this study, we repeatedly observed partial co-occurrence between mitochondrial mutations with other types of genomic alterations such as CNVs and nuclear SNVs nuclear mutations, suggesting the uniqueness of mtDNA variants in identifying subclones and its role as an additional fingerprint of crucial mutation events. With MQuad, it is now possible to take mtDNA variations into account during retrospective lineage tracing, hence a more sophisticated model for the integration of CNV, mtDNA and nuclear SNVs will be highly beneficial for clonal analysis and lineage reconstruction.

Additionally, other biological or technical factors should be considered to further enhance the effectiveness of MQuad in detecting clonally informed variants. For example, the strand specific allele information, like the strand correlation metric proposed in mgatk, may potentially filter out some low-quality variants presumably caused by technical reasons, hence worth further incorporation in a coherent way. Also, identifying variants that may be caused by post-transcriptional RNA editing may further enrich for variants arising from actual genetic variants or RNA editing events that are indeed (sub)clone specific. However, separating clone-specific RNA editing events and clonal genetic mutations is highly challenging. We anticipate that systematic characterization of RNA editing in a population scale can serve as a blacklist when detecting clonal mutations.

To conclude, the methods presented here unlock the untapped potential of existing single-cell sequencing data and provide an effective approach in lineage reconstruction with mtDNA variants alone or together with nuclear SNVs and/or CNVs. As new sequencing technologies are evolving, MQuad opens up a paradigm of analysis for a variety of single-cell sequencing datasets.

## Methods

**Variant detection pipeline**. The tailored variant detection pipeline can be briefly divided into 3 major steps:

Firstly, raw reads from BAM files are piled up using cellSNP-lite (v1.2.1)[23]. This generates a SNP-by-cell matrix in the form of a VCF file or sparse matrices of each cell's AD and DP at each variant position. The output includes every SNP found in the mitochondrial genome, which contains a large amount of noise and uninformative variants. MQuad (v0.1.6) takes this output and selects high quality informative variants with a binomial mixture model (explained in the next section). Lastly, vireoSNP (v0.5.3)[24] uses variational inference to reconstruct clonal populations based on the selected SNPs from MQuad.

**MQuad model**. MQuad assesses the heteroplasmy of mtDNA variants with a binomial mixture model. Compared to the Gaussian mixture model, the binomial mixture model considers the heteroplasmy as a proportional value and can directly exploit the raw read counts. In this model, it is assumed that the number of reads (or UMIs) for alternate allele (AD) for a SNP follow a binomial distribution with total trials as the depth of both alleles (DP) and success rate depending on the presence or absence of a variant:

$$P(AD|DP, \theta, I) = \begin{cases} Binom(AD|DP, \theta_0) \text{ for I = 0 (absence of variant)} \\ Binom(AD|DP, \theta_1) \text{ for I = 1 (presense of variant)} \end{cases} \quad (1)$$

Assuming there are $M$ cells in the sample, and the proportion of cells carrying a certain SNP is $\pi$, the likelihood can be estimated by:

$$L(\pi, \theta) = \prod_{j=1}^{M} \{Binom(AD_j|DP_j, \theta_1)\pi + Binom(AD_j|DP_j, \theta_0)(1 - \pi)\} \quad (2)$$

This likelihood of a 2-component binomial mixture model (M1) can be maximized using an expectation-maximization algorithm (pseudo code in Supplementary Algorithm 1) in order to get a maximum-likelihood estimation of $\pi$ and $\theta$. The same is done on a 1-component model (M0; i.e., $\pi = 0$) via direct maximization. For each fitted model, the Bayesian Information Criterion (BIC) can be calculated with the obtained likelihood ($L$) and the penalty on the number of parameters (1 for M0 versus 3 for M1) by

$$BIC = n_{parameters} \times \log(n_{cells}) - 2\log(L) \quad (3)$$

Consequently, the difference in Bayesian Information Criterion (ΔBIC) between models M1 and M0 can be calculated by ΔBIC = BIC (M0)—BIC(M1). The ΔBIC is further used as an indicator of clonal informativeness for each SNP with higher ΔBIC being more informative.

Finally, a ΔBIC cutoff for selection of SNPs is determined using the Kneedle algorithm[36], from a Python package kneed (v0.7.0). Briefly, Kneedle defines the curvature of any continuous function $f$ as a function of its first and second derivative:

$$K_f(x) = \frac{f''(x)}{\left(1 + f'(x)^2\right)^{1.5}} \qquad (4)$$

The algorithm aims to locate the 'knee' point by finding the point where $K_f(x)$ is maximum. This corresponds to our aim to find a point where the ΔBIC sharply increases to identify outlier SNPs which are most likely to be clonally discriminative.

**Data preprocessing**. The scRNA-seq data (Kim and fibroblast datasets; both Smart-seq2) was preprocessed largely based on the guidelines from GATK: (https://gatk.broadinstitute.org/hc/en-us/articles/360035535912-Data-pre-processing-for-variant-discovery). Specifically, FASTQ files were first aligned to the reference genome downloaded from UCSC (hg19 for human, mm10 for mouse) using STAR aligner (v2.7.2a)[37] with default parameters. Then, duplicates were marked and removed from the raw BAM files using MarkDuplicates from Picard (v2.18.9), resulting in analysis-ready BAM files for variant detection.

For 10x data (MKN-45 scDNA-seq & TNBC scRNA-seq), analysis read BAM files are directly downloaded and used. It should be noted that they are aligned to the hg38 reference, different from our processed data.

The mtscATAC-seq data (CRC dataset) was preprocessed with CellRanger-ATAC v1.2.0 with suggested modifications from the original paper. Briefly, NUMT regions from the standard hg19 reference were masked as suggested by Lareau et al.[16], followed by standard CellRanger-ATAC pipeline processing. We also explicitly removed frequent false positive variants occurring in error prone regions as described by Xu et al.[22], including rCRS 302–315, 513–525, 3105–3109.

**Simulation**. Mitochondria reads were synthesized in silico with NEAT-genReads v3.0[38] and SomatoSim (v1.0.0)[39]. The parameters for simulation were estimated from a primary haematopoiesis scRNA-seq dataset analyzed by Lareau et al.[16]. Briefly, pair-end aligned reads were generated for 90 simulated cells using NEAT. The empirical fragment length distribution was computed from the aforementioned dataset using the compute_fraglen.py function from NEAT. To simulate the uneven coverage observed in real datasets, positions with mean coverage higher than median were compiled up into a BED file, which was used in the -tr option in NEAT. These regions will have a coverage of 1000x while the off-target regions will only have 2% of the target coverage.

The simulation was performed based on the standard hg19 reference genome, with a read length of 152 bp. The mutation rate and error rate were set to 0 in this step as the simulator was not designed specifically for the mitochondrial genome hence not very flexible in simulating noisy variant allele frequencies.

To solve this problem, SomatoSim was used to simulate background noise instead. Firstly, we ran cellsnp-lite on the hematopoietic colony scRNA-seq dataset (536 cells) to get a cell-by-SNP matrix of VAF. Then, we selected variants that are 1) deeply covered with average depth >50 and 2) heteroplasmic (VAF > 1%) in more than 5% of the total cells, which totals up to around 3500 SNPs. Lastly, for each selected SNP, we randomly sampled the VAF to 90 cells, resulting in a smaller cell-by-SNP matrix that is used as simulation parameters.

For the simulation of clonal mutations, each clone has $M$ informative variants ($M \in \{5, 10, 25, 50\}$). The allele frequency of the clone-specific variants is sampled from a log-normal distribution of Lognormal($\mu$, 0.005) ($\mu \in \{0.01, 0.05, 0.1\}$). The clones are distributed in a ratio $r$ ($r \in \{1:1:1, 2:4:4, 2:2:6, 1:1:8\}$) and have a lineage tree $t$ ($t \in \{$'linear', 'mixed', 'branched'$\}$, Supplementary Fig. 1c). Each parameter was varied to simulate different contexts (Fig. 1e–h). When varying parameters, all other parameters were set to default ($M = 10$, $\mu = 0.1$, $r = 1:1:1$, $t = $ 'branched').

**Comparison with other tools**. We compared MQuad with mgatk on the simulated dataset with ground truth (results in Fig. 1b–h) and Kim dataset with tumor source labels (Fig. 2). We ran mgatk (v0.6.2) with default settings on both datasets. For the simulated dataset, we only varied the $\log_{10}(VMR)$ to call variants and obtained the PR and ROC curves in Fig. 1b–h, while the strand information is not used, as it is not supported by the simulator. On the Kim dataset, we used the suggested cutoffs to call variants: $\log_{10}(VMR) \geq -2$ and strand correlation coefficient $\geq 0.65$, which returns 312 variants.

For comparison with Monovar on the simulated dataset, we ran Monovar with recommended parameters. As Monovar was not designed specifically to only detect clonally informed variants, there is no direct metric that can be evaluated. However, the MPR field describes the 'Log Odds Ratio of maximum value of probability of observing non-ref allele to the probability of observing zero non-ref allele', which is the most relevant parameter in Monovar's output. PR and ROC curves were then obtained by varying the MPR statistic.

**Analysis of copy number variations on scDNA-seq data**. For the MKN-45 scDNA-seq dataset, we first explored the copy number variation profile by Cell-Ranger, a build-in software from 10x Genomics (Supplementary Fig. 3a). The CellRanger calling result shows a potentially interesting clonal structure with a unique small clone (node ID: 10050) carrying CN loss on chr4q, while a substantial fraction of cells and genomic regions may suffer from high error rate due to the high genomic variability in cancer cell line and the lack of allelic information. Therefore, we first confirmed that there is a genuine CN loss on chr4q in this group of cells by presenting its averaged B-allele frequency with comparison to the remaining cells (Supplementary Fig. 3b). In order to identify a cleaner clone assignment, we further used the read counts for both alleles on chr2 and chr4 to cluster cells into two clones with a binomial mixture model implemented in VireoSNP (Fig. 4a; Supplementary Fig. 3c). We noticed that with such BAF information, a two-cluster structure can be easily identified as suggested in the original study[29], which is also well concordant to the cellranger detected clusters (Supplementary Fig. 3d).

In order to visualize smoothed BAFs in single-cells, we used a three-level phasing strategy to aggregate multiple SNPs, as introduced in CHISEL[30]. First, allelic read counts were summed up for SNPs in a 50 Kb block by reference-based phasing with Sanger Imputation server. Second, we combined 100 blocks into a 5 Mb bin by a shared allelic ratio. Third, the B alleles in near bins were flipped using a dynamic programming algorithm to achieve a minimal BAF discrepancy with neighbor bins. It should be noted that procedures of the aggregation across blocks (step 2) and allele flipping across bins (step 3) only contribute to the visualization in Fig. 4a, but do not affect the clustering of cells into CNV clones as they were directly based on the aggregated SNPs in a 50 Kb window from reference-based phasing (step 1).

**Gene expression analysis**. Differential expression analysis on the fibroblast dataset was performed using the quasi-likelihood F-test function from edgeR[40]. To test for statistically significant differences in gene expression between clone1.MT0 and clone1.MT1, we fit a generalized linear model for single-cell gene expression with cellular detection rate, plate, and assigned mitochondrial clones as predictor variables.

Gene set enrichment analysis was performed using the camera function from limma[41,42]. Using 50 Hallmark gene sets from Molecular Signatures Database (MsigDB)[43], we tested for their enrichment using the log2-fold-change from the previous edgeR model as input.

Both DE analysis and gene set enrichment steps were adjusted for multiple testing by FDR estimation using independent hypothesis weighting from IHW[44]. The independent covariate used was average gene expression.

Differential expression analysis on the TNBC1 dataset was performed using the FindMarkers function from Seurat[45] using the DESeq2 model[46].

**Reporting summary**. Further information on research design is available in the Nature Research Reporting Summary linked to this article.

## Data availability

All datasets used are all publicly available, through the NCBI Gene Expression Omnibus (GEO) portal or ArrayExpress database at EMBL-EBI, with accession numbers GSE73121 (Kim dataset), GSE148673 (TNBC datasets), GSE148509 (CRC dataset), E-MTAB-7167 (Fibroblast dataset).

MKN-45 scDNA-seq and the melanoma 5′ scRNA-seq datasets, in bam format, are both downloaded from 10X Genomics website.

## Code availability

MQuad is an open-source Python package available at https://github.com/single-cell-genetics/MQuad. All the analysis notebooks and intermediate files for reproducing the results are available on https://doi.org/10.5281/zenodo.6054476.

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

## Acknowledgements

We thank Xianjie Huang for optimizing cellSNP-lite on genotyping mitoDNA variants and preprocessing MKN-45 data, and Yin Hei Lam for reproducing CopyKat results on TNBC1 data. This work was supported in part by AIR@InnoHK administered by Innovation and Technology Commission (J.H.), Collaborative Research Fund by the Research Grants Council of Hong Kong (C7026-18G, J.H.) and startup funds from the University of Hong Kong (Y.H.) and the Chinese University of Hong Kong (M.S.).

## Author contributions

J.H., Y.H., and M.S conceived and supervised the study. Y.H., J.H., A.K., C.Q. designed the statistical model. A.K. and C.Q. implemented the software. A.K. performed all data analysis with help from R.H. A.K., Y.H., and J.H. wrote the manuscript. All authors provided feedback on and approved the paper.

## Competing interests

The authors declare no competing interests.
