## [Peer Review File · Nature Communications]

Reviewers' Comments:

Reviewer #1:

Remarks to the Author:

The manuscript describes a computational method, MQuad for selecting informative variants from the noisy output of SNPs detected by existing tools to detect genetic clones in single cell sequencing data. While it is a quite useful tool in exploring an additional layer of omics information from current single cell sequencing datasets, there are several technical parameters that need to be further tested to better demonstrate its performance. My detailed comments are as follows:

1. Simulation – The more variants detected in each cell the easier it is to detect clones. The authors simulated 49-50 informative variants per clone. It is suggested to show the performance of MQuad in a range of simulated datasets with 1, 5, 10, 20 informative variants per clone.
2. MQuad aims to detect informative variants that can distinguish subpopulations by re-filtering the output of a published method, cellSNP-lite. While this method was designed to detect SNPs that distinguish subpopulations, it is suggested to add a function to detect mutations that are commonly shared across all clones to accurately define each clone.
3. MQuad uses binomial distribution to model alternative allele frequencies, followed by the Kneedle algorithm to find the cutoff of number of SNPs. What is the difference in detecting informative variants when different cutoffs are used? Do they significantly affect the robustness of informative variant selection?
4. There must be some errors in the speculated lineage tree in Figure 4d that was constructed from heatmap in Fig 4c. The 2116C>A is not unique to MT3 lineage; 7263 G>A and 15148 G>A are not unique to MT2; 12014C>T is not unique to MT4. Additionally, there are many shared mutations that are not shown in this tree.
5. The authors suggested to have 3K reads mapped to mtDNAs to apply MQuad. How about scRNA-seq data? What are the QCs to confidently apply MQuad to 3'- or 5'- scRNAseq datasets?
6. This is related to #4, could the authors test the performance of MQuad on 5'-scRNAseq datasets?

Reviewer #2:

Remarks to the Author:

The authors introduce a novel method, MQuad, for identifying mitochondrial variants from single-cell RNA or DNA sequencing data. MQuad performs a BIC-based model selection to infer the clonally informative mtDNA variants. The manuscript presents the application of the method on multiple single-cell datasets generated using different platforms. Using simulated data, the authors showed that MQuad performs better than other methods such as mgatk. MQuad+Vireo combination was also shown to have better clonal concordance as compared to mgatk+Vireo. Given the utility of mtDNA variants for in-vivo lineage tracing, variant calling from mtDNA is an important problem. MQuad and the proposed cellSNP-MQuad-VireoSNP pipeline seems to provide a good solution to the mtDNA variant calling problem. I however, have some questions/comments as described below.

1. The authors mentioned that variant callers that assumes a diploid context are not suitable for mtDNA variant calling due to varying heteroplasmy. However, this aspect was not explored well. What is the allele frequency distribution for mtDNA variants? Does MQuad perform equally well for variants of all VAF values? It is mentioned that clone-specific mtDNA variants not well-distinguishable from non-inherited mutations. MQuad does not perform any analysis for identifying whether a mutation is actually inherited or non-inherited. How its Bayesian model selection method perform in distinguishing between clonal vs non-inherited mutations. Can the authors design some experiments to explore these aspects?

2. For benchmarking MQuad, the authors used equal sized clones with 50 clone-specific mtDNA variants for each population. However, tumors often harbor more clones and the sizes of the clones also differ. The current benchmarking experiment does not tell us whether MQuad can recover clones of varying size both in terms of number of cells and mutations. The authors should design simulation experiments to explore these. Also, different tumors follow different models of evolution. Is MQuad equally capable of inferring the clone-specific mutations for linear, branching and neutral tumor evolution?

3. The authors compared the performance of MQuad only against mgatk. How does it perform against single-cell variant callers such as Monovar or Conbase or bulk sequencing caller such as GATK? Can the authors compare MQuad's performance against one such caller?

4. For the ccRCC dataset, the authors considered the three distinct sources as the three populations. However, each of these source populations could have harbored multiple clones. From the current analysis, it is clear that MQuad performs better in terms of assigning the cells to these 3 source populations. However, as the actual number of truly distinct populations could be more, it is not clear if MQuad can resolve that amount of heterogeneity. Can the authors perform some orthogonal analysis – such as utilizing tumor phylogeny inference methods like SCITE or SiCloneFIT on the called variants to infer the tumor phylogeny and compare its concordance with the MQuad+Vireo inferred tumor clones?

5. For the gastric cell line dataset, the authors uncovered 5 mtDNA clones. Given that the majority of these cells were associated with copy number clone 0, more discussion is required regarding the implication of the different mtDNA clones that are not distinguishable based on nuclear DNA.

Reviewer #3:

Remarks to the Author:

General

In this study, Kwok et al. present a new and fully probabilistic computational method for calling somatic variants from mitochondrial reads in single cell next generation sequencing data. There is a need for such new methods since previous approaches are based on heuristics and hard threshold values which are not guaranteed to allow for the discovery of low frequency and de-novo mtDNA variants. A key feature of MQuad is its use of Bayesian information criterion to distinguish mtDNA heteroplasmies informative for clonal assignment from those unrelated to clonality. The authors demonstrated improved clonal inference using MQuad in both simulated data and data with known ground truths, and compared results from MQuad primarily to an existing method mgatk, demonstrating in all cases MQuad performs better at inferring clonality. I have the following comments and questions.

Major

1. Informative SNPs: Can the authors present a plot of the allele frequency distribution of the non-informative and informative variants they detect using MQuad in the simulated and real datasets, so that the readers may have a intuitive understanding of the data used for clonal assignment?

2. Post transcriptional RNA-editing is known to be common in mitochondrial genes which violates this assumption. Can the authors comment how this affects identification of informative variants in MQuad?

3. Simulations: The simulation presented in the manuscript gave 49-50 informative variants, which seems a lot. Further, the authors are limited to 50% allele frequency in simulations because they have used NEAT-genReads which assumes a diploid context (from Figure 1b, it seems most of the informative variants assigned to each clone has an allele frequency of greater than 0.25). In real life datasets heteroplasmies would have occurred over a whole range, usually much lower than 50%. Though Figure 2a showed similarly high allele frequencies in a real dataset in all three cell populations, I would imagine this is only the case because a) only the top 20 informative SNPs were shown, and in fact the rest of the 133 informative SNPs identified showed much lower allele frequencies, and b) this is a tumour dataset which contains more mtDNA mutations than healthy tissue, as shown in Figure 3a. As such, I think it would be informative to perform simulations with

much lower allele frequencies at SNPs informative for clonality, over a range, to demonstrate effectiveness of MQuad in more realistic scenarios.

4. Comparison with mgatk: in addition to the variance mean ratio statistic, mgatk uses a strand correlation metric for identifying variants which are only found on one strand in the mtDNA cDNA library. These are filtered out since for most contemporary sequencing machines a strand specific photobleaching effect is known to exist in GC-rich regions which are common in mtDNA (see Schwarz 2011 and Lareau et al. 2020). Hence, it is a potential concern that some of the variants identified by MQuad are false positives - can the authors comment on this?

5. How are the authors excluding the possibility of contamination from nuclear mitochondrial sequences (NUMTs)? Please explain how the real data used in the manuscript are processed to reduce possibility of mismatched NUMT reads to the mtDNA reference and therefore showing up as a potential heteroplasmic SNP.

6. It would be great to include some runtime comparisons with mgatk and other softwares mentioned in the introduction of the paper, along with evaluation of sensitivity in identifying informative SNPs and clonal assignments.

Minor

1. Methods "MQuad Model": π is currently written as the proportion of cells carrying a certain SNP - is it not the proportion of cells being of a certain clone?

2. Methods "Gene expression analysis": What is the FDR threshold?

3. Results page 4: typo "inflection point" not "inflexion point"

4. Figure 1b: What do the authors mean by 'Only a small portion of background noise variants is shown for comparison with the actual clonal variants.' Do they exclude data from the heat map?

5. Figure 1c: From the caption it is not clear if the threshold was applied to true positive or true negative values.

6. Figure 1d: Is this the true CDF or the ECDF? There are several inflection points visible in the curve. What is the rationale behind choosing the first inflection point from the right?

7. Figure 3b: clone labels are misleading.

8. Figure 3d: have the log₁₀ P-values been shown in this plot been adjusted for multiple testing?

9. Please provide legends for all supplementary figures, tables and algorithms.

Reviewer 1: page 1-3

Reviewer 2: page 4-7

Reviewer 3: page 8-12

=====

Reviewer #1 (Expertise: clonal analysis of tumors, statistical methods for the analysis of single cell RNASeq analysis):

The manuscript describes a computational method, MQuad for selecting informative variants from the noisy output of SNPs detected by existing tools to detect genetic clones in single cell sequencing data. While it is a quite useful tool in exploring an additional layer of omics information from current single cell sequencing datasets, there are several technical parameters that need to be further tested to better demonstrate its performance. My detailed comments are as follows:

1. Simulation – The more variants detected in each cell the easier it is to detect clones. The authors simulated 49-50 informative variants per clone. It is suggested to show the performance of MQuad in a range of simulated datasets with 1, 5, 10, 20 informative variants per clone.

Response: Thank you for the suggestions. Now, we have tested the performance of MQuad with different numbers of clonal variants: 5, 10, 25, 50. In the new Fig. 1e, the number of informative variants per clone are varied and MQuad consistently outperforms other tools in all scenarios.

It should be noted that we have also re-designed the simulations to better mimic the experimental data (see the updated Methods, p.19-20).

2. MQuad aims to detect informative variants that can distinguish subpopulations by re-filtering the output of a published method, cellSNP-lite. While this method was designed to detect SNPs that distinguish subpopulations, it is suggested to add a function to detect mutations that are commonly shared across all clones to accurately define each clone.

Response: Thank you for suggesting the extra function to MQuad. Our understanding of your suggestion is to detect the heteroplasmy variants that occur in most cells but have different allele frequencies between clones. We agree that this is indeed a valuable feature for our method, and

we have added an additional column in the output table. This column flags variants that occur in most cells with varying and non-trivial heteroplasmy levels. In principle, these heteroplasmy variants are also clonally informed variants, so among the detected variants, we flag these heteroplasmy variants by setting thresholds on the minor component in MQuad output: proportion $\pi > 0.15$ and allele frequency $\theta > 0.1$.

3. *MQuad uses binomial distribution to model alternative allele frequencies, followed by the Kneedle algorithm to find the cutoff of number of SNPs. What is the difference in detecting informative variants when different cutoffs are used? Do they significantly affect the robustness of informative variant selection?*

Response: As the characteristics of mtDNA data vary between sequencing platforms and tissue types, it is generally difficult to set a universally applicable threshold on deltaBIC. Therefore, we employed the Kneedle algorithm to adaptively detect the cutoff. This cutoff will affect sensitivity and specificity, which is reflected in the PR and ROC curves in Fig. 1b-c. As shown by the black circle, the cutoff detected by Kneedle returns a good balance of sensitivity and specificity. On the other hand, a customised cutoff can also be used to give more weights to sensitivity or specificity. We have added analysis on the change in F1 score and MMC by varying the sensitivity parameter in Kneedle. In Supp. Fig. S2a, we showed that the F1 score and MMC in detecting informative variants stabilize at Sensitivity = 3 across all `n_informative_variants` scenarios, hence we think it is a good default value for robust selection of informative variants. Moreover, we also compared the clone assignment concordance when using different Kneedle parameters in the ccRCC dataset (Supp. Fig. S2b,c). It shows that the clonal assignment by vireoSNP is robust against the number of variants with the precision and recall stabilizing at $S = 3$. Nevertheless, user-defined Kneedle parameter can be specified when running MQuad as well.

4. *There must be some errors in the speculated lineage tree in Figure 4d that was constructed from heatmap in Fig 4c. The 2116C>A is not unique to MT3 lineage; 7263 G>A and 15148 G>A are not unique to MT2; 12014C>T is not unique to MT4. Additionally, there are many shared mutations that are not shown in this tree.*

Response: Thank you for pointing out this mistake. This was due to an overlooked version mismatch between the heatmap and the lineage tree. We have corrected it in the new Fig. 4d.

5. *The authors suggested to have 3K reads mapped to mtDNAs to apply MQuad. How about scRNA-seq data? What are the QCs to confidently apply MQuad to 3'- or 5'- scRNAseq datasets?*

Response: The 3'- or 5'- scRNA-seq data are indeed of high interest given their increasing popularity. In the original Fig. 5a, we compared the coverage of 3' scRNA-seq and SMART-seq2 data, and found that the 3' data gave significant bias on the coverage and missed a substantial proportion of mtDNA. That is probably why we only observed a small number of variants, which allows us to confidently assign only a small proportion of cells to clones in TNBC samples (see Supp. Table S1). We did not have the chance to evaluate the performance in 5' scRNA-seq in the original manuscript, but we suspect that it will suffer from its biased coverage. This is now further supported by additional analysis on one 5' data (see next point).

In general, the detection of mtDNA mutations in 3' or 5' biased scRNA-seq depends mostly on the position of mutations and whether they are well covered. In Supplementary Table S1, we showed the number of mitochondrial reads in each TNBC sample and the number of assignable cells. Even though there are a lot of mitochondrial reads in some of the samples, clonal mtDNA variants are not always found if they do not lie close to the deeply covered 3' ends. However, this can serve as a reference for the minimum QC for 10x datasets.

6. *This is related to #4, could the authors test the performance of MQuad on 5'-scRNAseq datasets?*

Response: We tested the performance of MQuad on a melanoma 5'-scRNAseq dataset publicly available on the 10x Genomics website. Though the total reads per cell (69.9K reads / cell) is lower in this 5' dataset compared to the 3' TNBC datasets in Supp. Table S1, it is still within a commonly used range (>50K reads / cell). Similarly with the 3' scRNA-seq dataset in Fig. 5, only a small number of variants can be detected by MQuad, and the clone assignment rate is also low (8%). Therefore, this observation well supports our statement that UMI-based platforms have more limited power in detecting mtDNA variants due to their biased coverage (also see previous point).

Reviewer #2 (Expertise: clonal analysis of cancer at the single cell level, computational and statistical biology):

The authors introduce a novel method, MQuad, for identifying mitochondrial variants from single-cell RNA or DNA sequencing data. MQuad performs a BIC-based model selection to infer the clonally informative mtDNA variants. The manuscript presents the application of the method on multiple single-cell datasets generated using different platforms. Using simulated data, the authors showed that MQuad performs better than other methods such as mgatk. MQuad+Vireo combination was also shown to have better clonal concordance as compared to mgatk+Vireo. Given the utility of mtDNA variants for in-vivo lineage tracing, variant calling from mtDNA is an important problem. MQuad and the proposed cellSNP-MQuad-VireoSNP pipeline seems to provide a good solution to the mtDNA variant calling problem. I however, have some questions/comments as described below.

1. The authors mentioned that variant callers that assume a diploid context are not suitable for mtDNA variant calling due to varying heteroplasmy. However, this aspect was not explored well. What is the allele frequency distribution for mtDNA variants? Does MQuad perform equally well for variants of all VAF values? It is mentioned that clone-specific mtDNA variants are not well-distinguishable from non-inherited mutations. MQuad does not perform any analysis for identifying whether a mutation is actually inherited or non-inherited. How its Bayesian model selection method perform in distinguishing between clonal vs non-inherited mutations? Can the authors design some experiments to explore these aspects?

Response: Thank you for the questions. We agreed that the suitability of diploid-based methods needs further exploration. Related to your point 3, we have compared MQuad against Monovar and found that it performs relatively poorly when compared to mtDNA-specific variant callers like MQuad and mgatk (Fig. 1). This is to be expected as Monovar is designed for diploid genomes to detect alternative alleles in any copy of the genome, hence resulting in a lot of false positive SNP calls in mtDNA. We also considered Conbase, which calls clonal mutations in the nuclear genome, but Conbase failed to run on chrM, possibly due to read phasing not available in the mitochondrial genome. Additionally, we presented the distribution of VAF values in Supp. Fig. S1a, showing the wide range of heteroplasmy values in mtDNA mutations. In our revised simulation, we also varied the VAF from 1% to 10%, showing that MQuad still outperforms other tools for other VAF values.

For your second question, our understanding of ‘non-inherited’ mutations is that they are either multi-cell spontaneous mutations that are not related to cell division dynamics, or germline mutations that happen in all the cells to start with. For the latter case, MQuad will filter them out as non-clonally-informed variants. For the former case, you are right that MQuad’s Bayesian model selection method does not directly distinguish between spontaneous mutations that happen in a subgroup of cells by chance and clonally propagated variants. However, the chance of a specific mutation independently occurring in more than 3 cells is close to 0 (10^{-9}) (Xu et. al, 2019; PMID: 30958261). MQuad also by default filters out variants that have a minor component of less than 2 cells as an additional safeguard against false positives.

2. For benchmarking MQuad, the authors used equal sized clones with 50 clone-specific mtDNA variants for each population. However, tumors often harbor more clones and the sizes of the clones also differ. The current benchmarking experiment does not tell us whether MQuad can recover clones of varying size both in terms of number of cells and mutations. The authors should design simulation experiments to explore these. Also, different tumors follow different models of evolution. Is MQuad equally capable of inferring the clone-specific mutations for linear, branching and neutral tumor evolution?

Response: Thank you for the very good suggestions. Before introducing these more systematic evaluations for MQuad, we first re-designed the simulations to better mimic the experimental data in non-diploid genomes (see the updated Methods, p.19-20).

As shown in the new Fig. 1 and Supp. Fig. S1, we have assessed the performance of MQuad when varying a few key parameters in the simulation, including different numbers of clonal variants: 5, 10, 25, 50; different clone size ratios; different models of evolution; and different allele frequency of the alternative allele. In all these settings, we found that MQuad consistently outperforms mgatk and Monovar, achieving a large gain in the AUPRC thanks to its good control of false positives. We also noticed that certain scenarios are more challenging for all methods, including extreme low allele frequency (AF=0.01; Fig. 1f) and linear structure compared to mixed and branched (Fig. 1h; Supp. Fig. S1c).

3. *The authors compared the performance of MQuad only against mgatk. How does it perform against single-cell variant callers such as Monovar or Conbase or bulk sequencing caller such as GATK? Can the authors compare MQuad's performance against one such caller?*

Response: It is indeed important to further compare with other variant callers even not designed for mtDNA variants. We have now added comparisons with Monovar in the simulation experiments (Fig. 1). Conbase was also considered but it failed to run due to the requirement of read phasing that is not available in chrM. In general, Monovar has good sensitivity, but it also returns a large number of false positives (Fig. 1b-c), probably because it is designed for diploid genomes, hence aiming to detect variants with alternative alleles in any copy of the genome, which is violated in mtDNA. We didn't include GATK here, as there is no direct statistic or score to use for calling the variant after it is performed per cell.

4. *For the ccRCC dataset, the authors considered the three distinct sources as the three populations. However, each of these source populations could have harbored multiple clones. From the current analysis, it is clear that MQuad performs better in terms of assigning the cells to these 3 source populations. However, as the actual number of truly distinct populations could be more, it is not clear if MQuad can resolve that amount of heterogeneity. Can the authors perform some orthogonal analysis – such as utilizing tumor phylogeny inference methods like SCITE or SiCloneFiT on the called variants to infer the tumor phylogeny and compare its concordance with the MQuad+Vireo inferred tumor clones?*

Response: Thank you for the suggestion. We agree that the actual number of distinct populations can be difficult to detect, particularly with a small number of cells. We have now added orthogonal analysis with SCITE on the ccRCC dataset, in Supplementary Fig. S3b. By visualizing the branch length in the hierarchical tree from SCITE, we did not see strong sub groups of cells. On the other hand, we found the structure of the cell source is only partially retained by SCITE, even though the same variant sets are used as for VireoSNP. This may be due to the limited power of model assumption in SCITE by only taking values of mutation presence or absence instead of allelic counts that are used in VireoSNP.

5. For the gastric cell line dataset, the authors uncovered 5 mtDNA clones. Given that the majority of these cells were associated with copy number clone 0, more discussion is required regarding the implication of the different mtDNA clones that are not distinguishable based on nuclear DNA.

Response: Thank you for the comment. We have now added more discussions in the Discussion section (p.17) on the potential co-occurrence or independence between different mutations and different molecular levels.

Reviewer #3 (Expertise: Variant calling in nuclear genomes):

General

In this study, Kwok et al. present a new and fully probabilistic computational method for calling somatic variants from mitochondrial reads in single cell next generation sequencing data. There is a need for such new methods since previous approaches are based on heuristics and hard threshold values which are not guaranteed to allow for the discovery of low frequency and de-novo mtDNA variants. A key feature of MQuad is its use of Bayesian information criterion to distinguish mtDNA heteroplasmy informative for clonal assignment from those unrelated to clonality. The authors demonstrated improved clonal inference using MQuad in both simulated data and data with known ground truths, and compared results from MQuad primarily to an existing method mgatk, demonstrating in all cases MQuad performs better at inferring clonality. I have the following comments and questions.

Major

1. Informative SNPs: Can the authors present a plot of the allele frequency distribution of the non-informative and informative variants they detect using MQuad in the simulated and real datasets, so that the readers may have an intuitive understanding of the data used for clonal assignment?

Response: Thank you for the suggestion. This plot is now added to Supp. Fig S1a, showing the allele frequency distribution of informative variants from simulated and real hematopoietic colonies scRNA-seq. As shown in this figure, there are generally two modes for clonal variants, where one with higher allele frequency represents the real mutation, and one with low frequency denotes background noise. N.B., we have also re-designed the simulations to better mimic the experimental data in non-diploid genomes (see the updated Methods, p.19-20; also the point 3 below)

2. Post transcriptional RNA-editing is known to be common in mitochondrial genes which violates this assumption. Can the authors comment how this affects identification of informative variants in MQuad?

Response: Thank you for the question. Post-transcriptional RNA editing is indeed a very relevant process for clonal variants detection. Assuming a uniform editing probability over all positions and

all cell types, MQuad model should be able to filter out these editing events as it should manifest as a one component binomial model. On the other hand, RNA-editing events may be unevenly distributed between cells (possibly in different states), which makes it hard for MQuad to distinguish between mutations arising post-transcriptional RNA-editing events from mutations from somatic mutation. We are expecting that in near future more systematic examinations of post-transcription of RNA editing events will be performed in a population scale with leveraging assays at both DNA and RNA levels. Then the catalog of such common RNA editing events can be flagged in MQuad for potential warnings.

However, if RNA-editing events are truly uneven (i.e. biased to some positions/some cell types), then it might be due to some biological differences and also give us insight to subpopulations nevertheless, which is in line with the purpose of clonal discovery. Nonetheless, we have added substantial discussion on this in the Discussion section (p.17).

3. Simulations: The simulation presented in the manuscript gave 49-50 informative variants, which seems a lot. Further, the authors are limited to 50% allele frequency in simulations because they have used NEAT-genReads which assumes a diploid context (from Figure 1b, it seems most of the informative variants assigned to each clone has an allele frequency of greater than 0.25). In real life datasets heteroplasmies would have occurred over a whole range, usually much lower than 50%. Though Figure 2a showed similarly high allele frequencies in a real dataset in all three cell populations, I would imagine this is only the case because a) only the top 20 informative SNPs were shown, and in fact the rest of the 133 informative SNPs identified showed much lower allele frequencies, and b) this is a tumour dataset which contains more mtDNA mutations than healthy tissue, as shown in Figure 3a. As such, I think it would be informative to perform simulations with much lower allele frequencies at SNPs informative for clonality, over a range, to demonstrate effectiveness of MQuad in more realistic scenarios.

Response: Thank you for the detailed suggestions. Related to the point 1 above, we have re-designed the simulations to better mimic the experimental data (see the updated Methods, p.19-20). Specifically, the clonal variants are now simulated at a much lower frequency (10% by default) with a fixed variance. Besides, we also varied the allele frequency to extreme low coverage to further examine the range of its applicability (Fig. 1f), along with a few other key simulation parameters. In all these settings, we found that MQuad consistently output performs mgatk and Monovar, with achieving a large gain in the AUPRC thanks to its well control of false positives.

4. Comparison with *mgatk*: in addition to the variance mean ratio statistic, *mgatk* uses a strand correlation metric for identifying variants which are only found on one strand in the mtDNA cDNA library. These are filtered out since for most contemporary sequencing machines a strand specific photobleaching effect is known to exist in GC-rich regions which are common in mtDNA (see Schwarz 2011 and Lareau et al. 2020). Hence, it is a potential concern that some of the variants identified by MQuad are false positives - can the authors comment on this?

Response: Thank you for the comments. The strand correlation metric (SCM) indeed has good potential in filtering low quality variants. In Lareau et al. 2020 (Fig. 3b), the low-quality and high-quality variants show distinct distributions on SCM. However, in our observations, this metric shows much less distinct two or multiple modes (Supp. Fig. S3a for Kim dataset), hence making it difficult to set a sensible threshold. Therefore, we did not include this metric in MQuad, but instead we have added more discussions in the Discussion section for its potential use in future (p.17-18).

5. How are the authors excluding the possibility of contamination from nuclear mitochondrial sequences (NUMTs)? Please explain how the real data used in the manuscript are processed to reduce possibility of mismatched NUMT reads to the mtDNA reference and therefore showing up as a potential heteroplasmic SNP.

Response: Thank you for the question. To avoid detecting false positive SNPs resulting from NUMT contamination, we first aligned the data to the standard hg19 reference genome, then cellSNP by default filters out reads with low MAPQ (< 20) and orphan (unpaired) reads. A similar strategy is also used in recent publications on somatic mutation detection (Park et. al, 2021; PMID: 34433967).

We also performed a simple simulation by sampling reads from the NUMT regions in the nuclear genome (hg19 blacklist by Lareau et. al 2021) with random mutations, then aligning to the whole hg19 reference using the same method, STAR. We found that among 4 million simulated reads, only 104 mapped to chrM, and mostly (90%) with very low MAPQ (< 5).

6. It would be great to include some runtime comparisons with mgatk and other softwares mentioned in the introduction of the paper, along with evaluation of sensitivity in identifying informative SNPs and clonal assignments.

Response: Thank you for the suggestion. We have now included comparisons with Monovar in our revised simulation. Runtime comparisons are also included in Supplementary Fig. S7. In small datasets, our cellsn-lite & MQuad together take less time compared to either Monovar or mgatk. In larger datasets from barcode-based platforms, cellsn-lite & MQuad is slower than mgatk probably due to the parameter optimization in fitting the MQuad model. Nevertheless, cellsn-lite & MQuad can finish within ~2h for 5,000 cells (with 10 CPU cores), which is probably acceptable for most cases, and the running time can be further reduced linearly by using more CPU cores.

Minor

1. Methods "MQuad Model": π is currently written as the proportion of cells carrying a certain SNP - is it not the proportion of cells being of a certain clone?

Thank you for the question. In the MQuad model, it is a per variant model, so π indeed refers to the proportion of cells carrying a certain SNP. Only in the VireoSNP step, the proportion of cells being of a certain clone is introduced. We have clarified this in the Methods section now (p.18-19).

2. Methods "Gene expression analysis": What is the FDR threshold?

The FDR thresholds are 10% for differential expression and 5% for gene set enrichment respectively, as shown in Fig. 3d and e.

3. Results page 4: typo "inflection point" not "inflexion point"

Thank you for pointing out the typo. It has been corrected now.

4. Figure 1b: What do the authors mean by 'Only a small portion of background noise variants is shown for comparison with the actual clonal variants.' Did they exclude data from the heat map?

Yes, they were excluded due to the figure size limit. Since there are more than 1,000 simulated noisy variants, we only randomly sampled 30 to show in the heatmap (Revised figure has been moved to Supp. Fig. S1b).

5. *Figure 1c: From the caption it is not clear if the threshold was applied to true positive or true negative values.*

In the ROC curve, the threshold was applied to calculate both true positive rate (sensitivity) and true negative rate (i.e., specificity & 1 - false positive rate).

6. *Figure 1d: Is this the true CDF or the ECDF? There are several inflection points visible in the curve. What is the rationale behind choosing the first inflection point from the right?*

This is the ECDF. The several inflection points are mainly because of the compression using log-scale. The first inflection point from the right is chosen because it is more likely to represent the difference between clonal mutations and noisy non-clonal mutations, while the other inflection points are most likely the difference between noisy non-clonal mutations and technical errors. In the revised method, we changed to not use the log scale as it is more intuitive (only 1 inflection point) and also has a better performance in most cases.

7. *Figure 3b: clone labels are misleading.*

Corrected with thanks.

8. *Figure 3d: have the log₁₀ P-values been shown in this plot been adjusted for multiple testing?*

No, the P-values shown on the y-axis are not adjusted. However, we did select the DE genes with adjusted p values and the color of the dots indicate the FDR after adjustments using IHW with logCPM supplied as the covariate.

9. *Please provide legends for all supplementary figures, tables and algorithms.*

Thank you. More details are provided in the legends now.

Reviewers' Comments:

Reviewer #1:

Remarks to the Author:

Thanks to the authors for having addressed part of my questions. I have some concern about the impact, i.e. MQuad is a small tool for filtering raw SNPs. The published tool, mgatk was designed for the same application. While simple hierarchical clustering can do the same work, MQuad did not show why it deserves this much effort to develop a new tool? There are still some errors in this version. Further, the authors demonstrated its application onto Smartseq2 data and 10X scDNAseq data. Smartseq2 is well known for its low throughput resulting significantly less usages in nowadays high throughput single cell sequencing fields, which probably will limit the application of MQuad. Additionally, 10X genomics has discontinued its scDNA-seq product due to low data quality and other issues, which again will limit MQuad application. My specific comments are as follows:

1. MQuad used a binomial model. Could the authors demonstrate how well the real data fit into this assumption comparing to other models, such as mixed Gaussian model or mixed negative binomial?
2. Could the authors apply a simple clustering method to select clonal informative mutations out of the raw mutations? I expect to observe subclusters of mutations that are quite clonal informative. Could authors use them for clonal assignment and compare to MQuad results?
3. In the ccRCC data, the authors benchmarked the clonal assignment results based on tissue source labels. Theoretically, patient-derived xenograft of metastatic tumor and metastatic tumor directly from the patients could have same mutation signatures. They don't necessarily have different clonality.
4. In the fibroblast dataset, MQuad detected 24 SNVs that are shown in Fig. 3a. However, from the heatmap, at least the half SNVs are not clonal informative SNVs because their mean allelic ratios are zeros, i.e. shown in white colors. Could the authors comment on these?
5. In line 183, the authors talked about 11196G>A is a clonal specific mutation and directed reader to Fig. 3a and Fig. S3. However, neither Fig.3a nor Fig.S3 showed this specific mutation. This same issue happened to another mutation, i.e, 2619A>T, talked in Line 184.
6. In line 293, the authors claimed several mutations are germline mutations. How are these mutations are defined as germline mutations, instead of early somatic mutations?
7. Fig.4d showed the lineage tree of mtDNA mutations. Although I pointed out the errors and authored claimed the errors were corrected. I still observe several obvious errors by comparing the heatmap of Fig. 4c and Fig. 4d. 1) mutation 3436G>A is absent from MT3 and MT4, which can't be a shared mutation of MT1, MT2, MT3, MT4. 2) neither 7236G>A nor 8940C>T is a shared mutation of MT2, MT3, MT4. 3) multiple mutations are shared across MT2, MT3 and MT4 are not shown in lineage tree, such as 4184T>C, 16286C>T...
8. Still related to the lineage tree. The frequencies of mtDNAs were used to define clones, however, the lineage trees only used the presence/absence of mutations to build the lineage. This dis-concordance was not explained by the authors.
9. In the cell line scDNAseq data, I recall that 10X genomics protocol started from nuclei isolation to perform sparse single cell copy number sequencing. Since mtDNAs should be washed away during nuclei isolation step, I am confused that why there are still so many mtDNAs detected in this dataset. Could the authors clarify or consult 10X genomics?
10. In line 251, it is suggested to use 3,000 mt reads per cell to apply MQuad on scDNAseq data. First of all, this suggestion was based on the down-sampling of only one dataset. Second, 10X genomics has discontinued the scDNAseq product, which will limit the application of MQuad onto scDNAseq data. Third, what is the suggestion of coverage depth of Smartseq2 dataset?

11. This happened in many panels that the figure legends covered the figure contents. Fig. S4a used wrong Figure key legends.

12. The run time is quite high in high throughput single cell data shown in Fig.S7. Any possibility of speed up the calculation?

Reviewer #2:

I thank the authors for carefully revising the manuscript based on my comments as well as other reviewers' comments. All my comments regarding the allele frequency distribution of the mtDNA variants, benchmarking of MQuad under different parameter settings have all been addressed. The revised manuscript now presents much better benchmarking on the simulated datasets. I still have a minor comment regarding the analysis of the ccRCC dataset.

1. For the ccRCC dataset, the authors ran SCITE to infer a phylogeny and the tree structure partially retains the cell source. The authors argue that this is a shortcoming of SCITE and might be caused by the limited power of the SCITE model that only uses presence or absence of a mutation. However, the tree structure inferred by SCITE may actually be hinting towards the fact that each population coming from the distinct sources may have further subpopulations. Some of these subpopulations may be shared by the distinct sources. The signature of this is observed in Fig. 2a also, there are a set of clonal mutations shared by all three sources, another set of mutations shared by two of the three sources and some mutations are source specific. Three sources can only be used as three populations if they are guaranteed to be homogeneous which is mostly improbable as these are cancer samples. Without resolving the subclonal structure of the source populations, it is difficult to treat the source populations as clones (i.e., homogeneous single population). Is there any bulk dataset associated with these samples that the authors can use for orthogonal validation? If not, I would suggest performing clustering using SCG (<https://www.nature.com/articles/nmeth.3867>) or BnpC (<https://academic.oup.com/bioinformatics/article/36/19/4854/5864024>) on all the cells from the three sources to infer the subpopulations and use them to evaluate the prediction by a tool. Subpopulations can also be inferred from the mutation tree inferred by SCITE by performing k-medoid clustering of cells based on the inferred mutation profiles.

Reviewer #3:

Remarks to the Author:

The authors have addressed my questions and concerns. I have no further concerns. I recommend acceptance.

Reviewer 1: page 3-11

Reviewer 2: page 12-14

=====

Reviewer #1 (Remarks to the Author):

Thanks to the authors for having addressed part of my questions. I have some concern about the impact, i.e. MQuad is a small tool for filtering raw SNPs. The published tool, mgatk was designed for the same application. While simple hierarchical clustering can do the same work, MQuad did not show why it deserves this much effort to develop a new tool? There are still some errors in this version. Further, the authors demonstrated its application onto Smartseq2 data and 10X scDNAseq data. Smartseq2 is well known for its low throughput resulting significantly less usages in nowadays high throughput single cell sequencing fields, which probably will limit the application of MQuad. Additionally, 10X genomics has discontinued its scDNA-seq product due to low data quality and other issues, which again will limit MQuad application. My specific comments are as follows:

Response:

Thank you for acknowledging that the early concerns have been addressed and for providing further comments. Firstly, we'd like to re-emphasise the importance of the mitochondrial variants calling in single-cells, and the grand challenges we are facing on this problem caused by various sources, including biological and technical stochasticity of allele detection and commonly low coverages especially in droplet-based platforms. As demonstrated in the simulated data (Fig. 1), we found that the existing methods, both mitochondrial specific or nuclear specific ones, return high error rates. We anticipate that an effective variant calling method like MQuad will make a strong impact on both scientific and translational communities. Secondly, MQuad is designed to be a near-universal method that can be applicable to a broad range of platforms, including SMART-seq2, droplet-based scRNA-seq, and scDNA-seq. Even though SMART-seq2 is less popular, it is still preferable over droplet-based sequencing protocols if one is interested in investigating the impact of somatic mutation on transcriptional phenotype, e.g., (McCarthy, et al, Nature Methods, 2020, PMID: 32203388) Similarly, droplet-based scDNA-seq received less attention partially because of its high cost, but they are actively used in academia, including a recently developed protocol ACT (Minussi et al. Nature 2021, PMID: 33762732).

Nonetheless, we agree with the reviewer that additional platforms can further demonstrate the wide applicability of MQuad. Therefore, in this revision, we have added one barcode-based single-cell ATAC-seq dataset generated by the mtscATAC-seq platform. Indeed, MQuad in general returned strong evidence (high Δ BIC) for clonal variants and identified a set of clonal variants with high consistency to the original report (see p.17 and the new Supp. Fig. S7). We have also updated the Supp. Table S1 to summarise all datasets we used in the manuscript, which can serve as reference for readers.

1. MQuad used a binomial model. Could the authors demonstrate how well the real data fit into this assumption comparing to other models, such as mixed Gaussian model or mixed negative binomial?

Response:

MQuad uses a binomial model, as our focus is the allele frequency in the cell population for each variant. Estimation of the allele frequency is non-trivial, as the total copies of molecules (or reads) are small, hence it can return a high sampling variability. Therefore, we chose a binomial model for such proportional variables, which is a common and preferred choice for modelling the allelic expression or coverage in single-cell datasets with low coverage.

Gaussian mixture model could be an option for modelling the allele frequency (AF) by taking the AD/DP values, but as mentioned above that this estimation of AF will suffer from sampling variability. Additionally, the Gaussian model is not bounded in the range between 0 and 1, hence may not be a sensible choice for a broad range of scenarios.

Negative binomial may be applicable for modelling the AD counts, by regressing on the depths DP, however there is no immediately available implementation for mixture of negative binomial regression models. On the other hand, this regression setting, specifically in the Poisson model (by eliminating the over dispersion term), approximates the binomial model asymptotically. Actually, we have implemented a beta-binomial mixture model to account for the potential over dispersion (data not shown in the paper). However, similar to the negative binomial model, the beta-binomial model also has no closed-form solution in the Maximization step, hence numerical optimization is needed, which takes much longer running time (>100x increase). As we didn't see

substantial improvement with sacrificing computing time by using the beta-binomial model, we used binomial mixture models in our manuscript throughout.

Overall, we hope that the reviewer will appreciate that our chosen model was tailored for this task in mtDNA variant detection.

2. Could the authors apply a simple clustering method to select clonal informative mutations out of the raw mutations? I expect to observe subclusters of mutations that are quite clonal informative. Could authors use them for clonal assignment and compare to MQuad results?

Response:

We have performed k-means clustering on all variants either by only using its average AF or together with the variant mean ratio (VMR) of the AF. Unsurprisingly, we didn't see a clean separation between the detected clonal variants and the background variants, no matter using MQuad or mgatk.

X denotes k-means centroids

*For mgatk, variants with strand_correlation < 0.65, mean coverage < 50, n_cells_conf_detected <= 5 were removed. For MQuad, variants with num_cells_minor_component <= 2 were removed

3. In the ccRCC data, the authors benchmarked the clonal assignment results based on tissue source labels. Theoretically, patient-derived xenograft of metastatic tumor and metastatic tumor directly from the patients could have same mutation signatures. They don't necessarily have different clonality.

Response:

Thanks for the comment. We agree that it is not necessary for PDX and patient mRCCs to be different clones, and we indeed see these two groups of cells are more similar compared to the primary RCC. On the other hand, MQuad can also detect the lineage informative variants, including those caused by random genetic drift that results in allele frequency (AF) shift between cell populations. For example, the variant 7527A>G has significantly different mean AF between PDX mRCC and Pt mRCC ($p=1.30e-5$, two-sided t-test).

Interestingly, the original study of this dataset (Supp. Fig. S1E in Kim *et al* 2016; PMID: 27139883; link below) reported that these three sources of cell population have substantially different copy number variations: Pt pRCC has chr5 copy gain uniquely; PDX mRCC has strong signal on copy number variations on chr12, 13, 14 but Pt mRCC is mostly neutral of these three chromosomes. Additionally, Poirion *et al*, (Fig. 6a-b; PMID: 30459309; link below) found that the source label has a high concordance with nuclear SNV based subclones. Therefore, it is likely to be true that distinct genetic divergence, including on mitochondrial genome, exists between these three source populations.

* Supp. Fig. S1E in Kim *et al*:

https://static-content.springer.com/esm/art%3A10.1186%2Fs13059-016-0945-9/MediaObjects/13059_2016_945_MOESM1_ESM.pdf

* Fig. 6a-b in Poirion *et al*: <https://www.nature.com/articles/s41467-018-07170-5/figures/6>

4. In the fibroblast dataset, MQuad detected 24 SNVs that are shown in Fig. 3a. However, from the heatmap, at least the half SNPs are not clonal informative SNVs because their mean allelic ratios are zeros, i.e. shown in white colors. Could the authors comment on these?

Response:

Thanks for pointing this out. The seemingly white color in the original Fig. 3a actually does not indicate zero allele frequency but a very small value. We have now updated the color scheme to better show the non-zero mean allelic ratios (Figure also attached below).

Old version (left) and new version (right)

5. In line 183, the authors talked about 11196G>A is a clonal specific mutation and directed reader to Fig. 3a and Fig. S3. However, neither Fig.3a nor Fig.S3 showed this specific mutation. This same issue happened to another mutation, i.e, 2619A>T, talked in Line 184.

Response:

Thanks for pointing out this mistake. The mentioned figure should be Fig. S4 instead of S3. It has been fixed now.

6. In line 293, the authors claimed several mutations are germline mutations. How are these mutations are defined as germline mutations, instead of early somatic mutations?

Response:

Thanks for the question. Indeed we cannot concretely distinguish if these are germline mutations or early somatic mutations, as we do not have a matched normal tissue for comparison. In this revision, we have rephrased it to “common mutations (either germline or early somatic mutations)”.

7. Fig.4d showed the lineage tree of mtDNA mutations. Although I pointed out the errors and authored claimed the errors were corrected. I still observe several obvious errors by comparing the heatmap of Fig. 4c and Fig. 4d. 1) mutation 3436G>A is absent from MT3 and MT4, which can't be a shared mutation of MT1, MT2, MT3, MT4. 2) neither 7236G>A nor 8940C>T is a shared mutation of MT2, MT3, MT4. 3) multiple mutations are shared across MT2, MT3 and MT4 are not shown in lineage tree, such as 4184T>C, 16286C>T...

Response:

Thanks for the comments again, and apologise for the lack of clarity in the color scheme. Same as the point 4, these seemingly white colors actually denote low AF values instead of zeros. We have now revised the color scheme in Fig. 4c (also attached below) and hope it clarifies now. Also, we have added the shared mutations to the root of the lineage tree.

Old version (left) and new version (right)

8. *Still related to the lineage tree. The frequencies of mtDNAs were used to define clones, however, the lineage trees only used the presence/absence of mutations to build the lineage. This dis-concordance was not explained by the authors.*

Response:

Indeed, we used the presence and absence of the mutation to build the lineage (both manually and with SCITE). We agree that this is indeed an open challenge and we have highlighted the reason accordingly (p.12)

9. *In the cell line scDNAseq data, I recall that 10X genomics protocol started from nuclei isolation to perform sparse single cell copy number sequencing. Since mtDNAs should be washed away during nuclei isolation step, I am confused that why there are still so many mtDNAs detected in this dataset. Could the authors clarify or consult 10X genomics?*

Response:

Thanks for your question. Indeed we observed a substantial amount of mitochondrial reads in all other 10x CNV datasets, varying from 2000 to 8000 MT reads per cell, showing it is not a dataset-specific problem, eg. Melanoma COLO829 (6k MT reads per cell) and Breast cancer Section E (2k MT reads per cell). After consulting with 10X genomics, their explanation is as follows:

“Most likely the presence of MT reads in the sample has to do with how the lysis step is performed during the nuclei isolation protocol. Lysis of the cell membrane can also cause lysis of the mitochondrial membrane, and the ambient mitochondrial genes could be sticking to the nuclei or being incorporated into the GEMs.”

We hope this clarifies any confusion regarding the dataset.

10. *In line 251, it is suggested to use 3,000 mt reads per cell to apply MQuad on scDNAseq data. First of all, this suggestion was based on the down-sampling of only one dataset. Second, 10X genomics has discontinued the scDNAseq product, which will limit the application of MQuad onto scDNAseq data. Third, what is the suggestion of coverage depth of Smartseq2 dataset?*

Response:

Thanks for your questions. Firstly, indeed we only down-sampled one dataset, because this particular dataset has relatively high coverage compared to others, hence allowing us to explore the impact of coverage on variant detection over a large range. Secondly, the discontinuation of 10X CNV product is indeed unfortunate, but the general droplet-based scDNA-seq technology is still active in the academia, eg. a recently developed protocol ACT (Minussi et al. Nature 2021, PMID: 33762732). Thirdly, Smart-seq2 datasets generally require 500K to 1M total reads per cell for saturated gene expression, which is sufficient coverage for using MQuad. For example, in the fibroblast dataset, there are 618K reads per cell (with 35K MT reads per cell). In short, most Smart-seq2 datasets should be suitable for this analysis.

In this revision, we have updated the Supp. Table S1 to summarise all datasets we used in the manuscript, which can serve as reference for readers on choosing the coverage.

11. This happened in many panels that the figure legends covered the figure contents. Fig. S4a used wrong Figure key legends.

Response: Thanks for raising the potential issues. In this revision, we have updated the legend location in Fig. 1b and 5a to avoid covering the figure contents.

12. The run time is quite high in high throughput single cell data shown in Fig.S7. Any possibility of speed up the calculation?

Response:

Indeed the running time of MQuad is higher than mgatk when large number of cells are in use (including both TNBC1 and MKN45 datasets). This is because more iterations are required when optimising the parameters in the binomial mixture model. On the other hand, we think it is generally acceptable to run less than 2 hours on >5,000 cells. Nonetheless, for a very large number of cells, we may consider potential ways for speedups while slightly sacrificing accuracy, e.g., by using mini-batch cells. We will oversee this through users' feedback and consider introducing new features for speedup in future releases.

Reviewer #2 (Remarks to the Author):

I thank the authors for carefully revising the manuscript based on my comments as well as other reviewers' comments. All my comments regarding the allele frequency distribution of the mtDNA variants, benchmarking of MQuad under different parameter settings have all been addressed. The revised manuscript now presents much better benchmarking on the simulated datasets. I still have a minor comment regarding the analysis of the ccRCC dataset.

*1. For the ccRCC dataset, the authors ran SCITE to infer a phylogeny and the tree structure partially retains the cell source. The authors argue that this is a shortcoming of SCITE and might be caused by the limited power of the SCITE model that only uses presence or absence of a mutation. However, the tree structure inferred by SCITE may actually be hinting towards the fact that each population coming from the distinct sources may have further subpopulations. Some of these subpopulations may be shared by the distinct sources. The signature of this is observed in Fig. 2a also, there are a set of clonal mutations shared by all three sources, another set of mutations shared by two of the three sources and some mutations are source specific. Three sources can only be used as three populations if they are guaranteed to be homogeneous which is mostly improbable as these are cancer samples. Without resolving the subclonal structure of the source populations, it is difficult to treat the source populations as clones (i.e., homogeneous single population). Is there any bulk dataset associated with these samples that the authors can use for orthogonal validation? If not, I would suggest performing clustering using SCG (<https://www.nature.com/articles/nmeth.3867>) or BnpC (<https://academic.oup.com/bioinformatics/article/36/19/4854/5864024>) on all the cells from the three sources to infer the subpopulations and use them to evaluate the prediction by a tool. Subpopulations can also be inferred from the mutation tree inferred by SCITE by performing *k*-medoid clustering of cells based on the inferred mutation profiles.*

Response:

We thank the reviewer for appreciating our revision. For this remaining comment on ccRCC, we agree that the source of cell populations may only represent a certain aspect for the heterogeneity among the cells, and subclonal structure may exist within or between cell source populations. On the other hand, multiple evidences have been reported that the source labels retain high divergence of somatic mutations, e.g., copy number variations (Supp. Fig. S1E in Kim *et al* 2016;

PMID: 27139883; link below) and nuclear SNVs (Fig. 6a-b in Poirion et al 2018; PMID: 30459309; link below). Therefore, even if subclones may exist within each of the cell source populations, the source label could function as a good annotation of clonality for certain (probably major) mutations.

* Supp. Fig. S1E in Kim *et al*:

https://static-content.springer.com/esm/art%3A10.1186%2Fs13059-016-0945-9/MediaObjects/13059_2016_945_MOESM1_ESM.pdf

* Fig. 6a-b in Poirion et al: <https://www.nature.com/articles/s41467-018-07170-5/figures/6>

Nonetheless, we thank the reviewer for suggesting the additional tools to identify more detailed genetic clusters of these cells from mtDNA SNVs. Here, we have performed both BnpC and SCG on these 146 mtDNA SNVs detected by MQuad. Similar to SCITE, we did not find a strong concordance between their clusters and either the source label nor the clusters identified by Vireo (Figure below). This is particularly concerning, as the source label has been shown as a major contributor to CNV clones, nuclear SNV clones and mtDNA clones (by Vireo). Therefore, we tend not to trust too much on BnpC and SCG, as they are developed for analysing nuclear SNVs based on scDNA-seq data by only using presence and absence of mutations, which may easily fail in capturing the heteroplasmy of mtDNA allelic fractions, particularly random genetic drifting.

Overall, we hope the reviewer agrees with us that the source label is a reasonable (despite coarse) annotation for benchmarking and it is an open challenge to infer a fine-grained lineage from mtDNA mutations (e.g., in a tree structure) as it presents different distributions compared to commonly studied nuclear SNVs.

\

Reviewers' Comments:

Reviewer #1:

Remarks to the Author:

Thank you for spending efforts in addressing my comments. I think that all of my previous comments have been addressed in the point by point response letter. Please add these responses into to the manuscript accordingly, either in related sessions or in discussion, particular I failed to locate the responses to 1,2,3,6 in this new version of manuscript.

One minor comment: In Figure 5A, please label the sequencing platform of TNBC1, 2, 5 samples

Reviewer #2:

Remarks to the Author:

I thank the authors for performing more analysis using BnPC and SCG on the ccRCC dataset. Given the discordance of the clonal populations inferred by the two tools, I agree with the authors that for this dataset, using the source labels for benchmarking is a reasonable choice given their differences in copy numbers and SNVs. I don't have any further comments and I recommend acceptance. The authors however need to fix a typo in Fig. 2 of the revised manuscript.

Fig. 2d, the title shows 'MQuad-VireoSNP' prediction. I believe it should be 'mgatk-VireoSNP' prediction.

RESPONSE TO REVIEWERS' COMMENTS

Reviewer #1 (Remarks to the Author):

Thank you for spending efforts in addressing my comments. I think that all of my previous comments have been addressed in the point by point response letter. Please add these responses into the manuscript accordingly, either in related sessions or in discussion, particular I failed to locate the responses to 1,2,3,6 in this new version of manuscript.

One minor comment: In Figure 5A, please label the sequencing platform of TNBC1, 2, 5 samples

Response: Thanks for your comment.

The response for points 3 and 6 had already been added in the previous revision p.8 (the second last paragraph) and p. 12, respectively. In this new revision, we have further added a short discussion *for point 1* in Methods section (p. 19) and brief highlight *for point 2* in p.5. We appreciate the reviewer's insistence on adding more discussion, while we prefer to have a balance between details and concision. Therefore, we keep some discussion brief, with referring to the original studies, e.g., for point 3 above.

The sequencing platforms are labelled in Fig. 5A now.

Reviewer #2 (Remarks to the Author):

I thank the authors for performing more analysis using BnPC and SCG on the ccRCC dataset. Given the discordance of the clonal populations inferred by the two tools, I agree with the authors that for this dataset, using the source labels for benchmarking is a reasonable choice given their differences in copy numbers and SNVs. I don't have any further comments and I recommend acceptance. The authors however need to fix a typo in Fig. 2 of the revised manuscript.

Fig. 2d, the title shows 'MQuad-VireoSNP' prediction. I believe it should be 'mgatk-VireoSNP' prediction.

Response: Thanks for pointing out the error. The typo in Fig. 2 has been fixed now.